# A new insight into the vertical differences in $NO_2$ heterogeneous reaction to produce HONO over inland and marginal seas

Chengzhi Xing[1], Shiqi Xu[7], Yuhang Song[2], Cheng Liu[2,1,3,4,*], Yuhan Liu[6], Keding Lu[5,*], Wei Tan[1], Chengxin Zhang[2], Qihou Hu[1], Shanshan Wang[9], Hongyu Wu[8], and Hua Lin[8]

[1] Key Lab of Environmental Optics & Technology, Anhui Institute of Optics and Fine Mechanics, Hefei Institutes of Physical Science, Chinese Academy of Sciences, Hefei 230031, China

[2] Department of Precision Machinery and Precision Instrumentation, University of Science and Technology of China, Hefei, 230026, China

[3] Center for Excellence in Regional Atmospheric Environment, Institute of Urban Environment, Chinese Academy of Sciences, Xiamen, 361021, China

[4] Key Laboratory of Precision Scientific Instrumentation of Anhui Higher Education Institutes, University of Science and Technology of China, Hefei, 230026, China

[5] State Key Joint Laboratory of Environment Simulation and Pollution Control, College of Environmental Sciences and Engineering, Peking University, Beijing 100871, China

[6] Department of unclear safety, Chia Institute of Atomic Energy, Beijing, 102413, China

[7] School of Earth and Space Sciences, University of Science and Technology of China, Hefei, 230026, China.

[8] School of Environmental Science and Optoelectronic Technology, University of Science and Technology of China, Hefei, 230026, China

[9] Shanghai Key Laboratory of Atmospheric Particle Pollution and Prevention (LAP[3]), Department of Environmental Science and Engineering, Fudan University, Shanghai, 200433, China

*Corresponding authors: Cheng Liu (chliu81@ustc.edu.cn); Keding Lu (k.lu@pku.edu.cn)

**ABSTRACT**
Ship based multi-axis differential optical absorption spectroscopy (MAX-DOAS) measurements were conducted
along the marginal seas of China from 19 April to 16 May 2018 to measure the vertical profiles of aerosol, nitrogen
dioxide ($NO_2$), and nitrous acid (HONO). Along the cruise route, we found five hot spots with enhanced
tropospheric $NO_2$ VCDs in Yangtze River Delta, Taiwan straits, Guangzhou-Hong Kong-Macao Greater Bay areas,
Zhanjiang Port, and Qingdao port. Enhanced HONO concentrations could usually be observed under high-level
aerosol and $NO_2$ conditions, whereas the reverse was not always the case. To understand the impacts of relative
humidity (RH), temperature, and aerosol on the heterogeneous reaction of $NO_2$ to form HONO in different scenes, the
Chinese Academy of Meteorological Sciences (CAMS) and Southern University of Science and Technology (SUST)
MAX-DOAS stations were selected as the inland and coastal cases, respectively. The RH turning points in CAMS and
SUST cases were both ~65% (60–70%), whereas two turning peaks (~60% and ~85%) of RH were found in the sea
cases. As temperature increased, the HONO/$NO_2$ ratio decreased with peak values appearing at ~12.5℃ in CAMS,
whereas the HONO/$NO_2$ gradually increased and reached peak values at ~31.5℃ in SUST. In the sea case, when the
temperature exceeded 18.0℃, the HONO/$NO_2$ ratio rose with increasing temperature and achieved its peak at ~25.0℃.
This indicated that high temperature can contribute to the secondary formation of HONO in the sea atmosphere. In the
inland case, the correlation analysis between HONO and aerosol in the near-surface layer showed that the ground
surface is more crucial to the formation of HONO via the heterogeneous reaction of $NO_2$; however, in the coastal and
sea cases, the aerosol surface contributed more. Furthermore, we discovered that the conversion rate of $NO_2$ to HONO
through heterogeneous reaction in the sea case is larger than that in the inland case in higher atmospheric layers (>
600 m). Three typical events were selected to demonstrate three potential contributing factors of HONO production
under marine conditions (i.e., transport, $NO_2$ heterogeneous reaction, and unknown HONO source). This study
elucidates the sea-land and vertical differences in the forming mechanism of HONO via the $NO_2$ heterogeneous
reaction and provides deep insights into tropospheric HONO distribution, transforming process, and environmental
effects.

**1 Introduction**
Nitrous acid (HONO) is an important part of the atmospheric nitrogen cycle and plays a significant role in
atmospheric oxidation capacity (Alicke et al., 2003; Kleffmann et al., 2005). Photolysis of HONO in near ultraviolet
bands (Eq. 1) is a substantial source of hydroxyl radicals (OH radicals), which are one of the most important oxidants
in the tropospheric atmosphere. Earlier studies reported that the contribution of HONO photolysis to OH radicals can

reach 40–60%, while exceeding 80% in the early morning (Michoud et al., 2012; Ryan et al., 2018; Xue et al., 2020). OH radicals can oxidize and destroy most atmospheric pollutants, such as CO, $NO_x$ (NO+$NO_2$), $SO_2$, and volatile organic compounds (VOCs), thereby further promoting the formation of secondary pollutants (e.g., ozone ($O_3$), peroxyacetyl nitrate (PAN), and secondary aerosols) and leading to serious haze pollution events (Huang et al., 2014). Additionally, as a nitrosating agent, HONO can produce carcinogenic nitrite amines that pose a threat to human health (Zhang et al., 2015). Therefore, a full understanding of the source and formation mechanism of HONO is scientifically significant for the study of tropospheric oxidation and the control of secondary pollution.

Currently, the known sources of HONO mainly include direct emissions from vehicles, ships, biomass burning and soil, the homogeneous reaction of NO and OH radicals (Eq. 2), the nighttime and daytime heterogeneous reaction of $NO_2$ (Eq. 3) on aerosols, vegetation, ground and other types of surfaces, and the photolysis of nitrate particles (Eq. 4) (Alicke et al., 2003; Stemmer et al., 2006; Indarto et al., 2012; Wang et al., 2015; Salgado and Rossi, 2002; Zhou et al., 2011). Sources of HONO exist that are poorly understood (Fu et al., 2019). The heterogeneous reaction of $NO_2$ as a source of HONO has received continuous attention in recent years. It was found that the heterogeneous reaction of $NO_2$ is one of the most important sources of HONO in a variety of scenes such as inland, coastal cities, and offshore seas. Liu et al. (2021) reported the contribution of heterogeneous reaction of $NO_2$ on aerosol surface to HONO is 19.2% in summer, and this contribution on aerosol and ground surfaces to HONO can reach 54.6% in winter in Beijing. Yang et al. (2021) and Zha et al. (2014) found that the generation rate of HONO through the heterogeneous reaction of $NO_2$ under sea-wind conditions could elevate 3–4 times than that under land-wind conditions in the northern coastal city of Qingdao and the southern coastal city of Hong Kong, respectively. Cui et al. (2019) illustrated that the heterogeneous reaction of $NO_2$ on aerosol and sea surfaces is an important source of HONO in East China Sea in summer. The process of HONO formed from the heterogeneous reaction of $NO_2$ is affected by various atmospheric parameters. The relative humidity (RH), temperature, solar radiation intensity (SRI), and aerosol concentration and its relative surface area are the particularly important parameters. Earlier works always used the linear regression relationship between HONO/$NO_2$ and the above parameters to characterize the influence of these parameters on the formation of HONO through the heterogeneous reaction of $NO_2$. Although this kind of simple linear regression method may lead to artificial correlations and misleading conclusions, considering the vertical evolution of atmospheric parameters. Wen et al. (2019) found that the increased temperature could promote the heterogeneous reaction of $NO_2$ to form HONO in sea conditions. The generation rate of HONO could increase rapidly, when the temperature was greater than 20 ℃. Gil et al. (2019) found that the HONO formed from the heterogeneous reaction of $NO_2$ will increase along with the increase of RH when RH was less than 80% in a case of land park using deep learning forced by measurement results. Fu et al. (2019) reported that RH and SRI were the main parameters driving the heterogeneous reaction of $NO_2$ to form HONO in Pearl River Delta, and it contributes to 72% of the total source of HONO. Cui et al. (2019) found that the potential of heterogeneous reaction of $NO_2$ to form HONO will increase with the increase of particle concentration and the specific surface area of single particle in coastal cities.

$$HONO + hv \rightarrow NO + \cdot OH \, (\lambda < 400nm) \tag{1}$$

$$\cdot OH + NO + M \rightarrow HONO + M \tag{2}$$

$$2NO_2 + H_2O \rightarrow HONO + HNO_3 \tag{3}$$

$$HNO_3 / NO_3^- + hv \rightarrow HONO / NO_2^- + O \cdot (\lambda \sim 300nm) \tag{4}$$

However, earlier researches generally focused on the near-surface layer of a single scene, and attentions to the influence mechanism of the heterogeneous reaction of $NO_2$ to form HONO in vertical direction and in different sea and land scenes are insufficient, which limits the comprehensive assessment to understand the sea-land differences and impact mechanism of HONO formed from the heterogeneous reaction of $NO_2$. $NO_2$ could be transported from inland and coastal cities to offshore seas (Tan et al., 2018). This part of $NO_2$ can promote the HONO formation through heterogeneous reaction on the high-level aerosol and sea surfaces in the atmosphere of sea (Zhang et al., 2020). The formed HONO is likely to be carried to land cities at night by sea breeze, which will affect the atmospheric oxidation and air quality, and even endanger human health. Additionally, the vertical distributions and values of atmospheric meteorology and aerosol parameters are significantly different in land and sea scenes, which provide different conditions for the heterogeneous reaction of $NO_2$ to form HONO in different height layers. Furthermore, aerosols and $NO_2$ have complex evolution and transmission characteristics in the vertical direction. The vertical upward transport of aerosol and $NO_2$ can promote the HONO formation through heterogeneous reaction at high altitude, and the vertical downward transport of HONO will impact the atmospheric environment near the ground. The vertical observations in land-sea scenes are also helpful to distinguish the contribution of the heterogeneous reaction of $NO_2$ on the aerosol and ground/sea surfaces (Zhang et al., 2020).

Currently, a variety of HONO measurement techniques have been developed, which in principle can be roughly divided into wet chemical, spectroscopy, and mass spectrometry methods (Cheng et al., 2013; Bernard et al., 2016; Gil et al., 2019; Guo et al., 2020; Jordan et al., 2020). However, these technical methods can only measure the HONO

information near the surface layer. Taking tower and aircraft as platforms, these techniques were performed to
measure HONO vertical profiles, and it was found that the peak values of HONO usually appeared under 200 m at
urban and suburban areas (Kleffmann et al., 2003; Stemmler et al., 2006; Zhang et al., 2009; Wong et al., 2012; Meng
et al., 2020; Zhang et al., 2020). These studies also revealed that the heterogeneous reaction of $NO_2$ on multiple
surfaces (ground and aerosol etc.) was an important source of HONO under planetary boundary layer (PBL),
especially in haze days. Furthermore, they also reported that the $HONO/NO_2$ ratios usually decreased with the
increase of height under 200 m at inland and coastal areas. However, the cost of above techniques used to measure
HONO vertical profiles was too high, and the real-time and continuous measurement cannot be realized. Multi-axis
differential optical absorption spectroscopy (MAX-DOAS), as a ground-based ultra-hyperspectral remote sensing
technology, was widely used for vertical observation of atmospheric pollutants in the past two decades. In the past
five years, several researchers carried out campaigns based on MAX-DOAS to measure the vertical profile of HONO
in inland and coastal areas, and revealed their vertical characteristics, sources, and the contribution to atmospheric
oxidation at different height layers (Garcia-Nieto et al., 2018; Ryan et al., 2018; Wang et al., 2020; Xing et al., 2021;
Xu et al., 2021; He et al., 2023). Few studies were conducted on the sources of HONO at different height layers in sea
conditions. In this study, MAX-DOAS is used for the first time to study the spatiotemporal distribution and the
sources of HONO along the Chinese coastline, and to learn the differences of the HONO formed from the
heterogeneous reaction of $NO_2$ in different height layers and land-sea scenes.

## 2 Methods and methodologies

### 2.1 Measurement cruise

The ship-based atmospheric observation campaign along the marginal seas of China was conducted from 19 April to
16 May 2018. The latitude and longitude ranges of the entire campaign covered $21.12^oN–35.89^oN$ and
$110.67^oE–122.16^oE$. The detailed voyage records of the observation ship are shown in Table 1. An integrated and
fully automated MAX-DOAS instrument was installed aboard the stern deck of the ship (Figure S1(a)). To ensure that
the instrument is always kept in a horizontal position, a photoelectric gyro was used. The angle between the
observation and heading directions of the ship was always maintained at $135^o$ during the whole campaign. The
telescope unit of the instrument pointed towards sea during cruise NO.3 and NO.6. The telescope unit pointed towards
inland during cruise NO.1, NO.4, and NO.5. During cruise NO.2, the observation telescope always pointed to
Chongming island. The measurement ship only sailed in daytime from 19 April to 02 May, and continuously sailed in
all the daytime and nighttime from 3 May to 16 May 2018. The ship docked in Daishan port on 9–10 May and no
observations were conducted during these two days.
The aim of this campaign was to learn the vertical differences of $NO_2$ heterogeneous reaction to produce HONO in
marginal seas of China and compare the influence mechanism of that in inland cities. To fully understand the
differences of the impacts of RH, temperature, and aerosol on the HONO secondary formation in land and sea
conditions, the Chinese Academy of Meteorological Sciences (CAMS) and Southern University of Science and
Technology (SUST) MAX-DOAS stations were selected as inland and coastal areas for analysis, respectively. CAMS
is located in the urban of Beijing ($116.32^oE$, $39.94^oN$), and SUST is located in Shenzhen ($114.00^oE$, $22.60^oN$) (Figure
S2). This study will provide scientific guidance for understanding regional oxidation capacity and controlling the
secondary air pollution.

### 2.2 MAX-DOAS measurements

#### 2.2.1 Instrument setup

The compact instrument consists of an ultraviolet spectrometer (AvaSpec-ULS2048L-USB2, 300–460 nm spectral
range, 0.6 nm spectral resolution) at a 20℃ fixed temperature with a deviation of $< 0.01$℃, a one-dimensional CCD
detector (Sony ILX511, 2048 individual pixels) and a telescope unit driven by a stepper motor to collect scattered
sunlight from different elevation angles. The accuracy of elevation angle is $< 0.1^o$ and the telescope field of view
(open angle) is $< 0.3^o$. A full scanning sequence consists of 11 elevation angles ($1^o$, $2^o$, $3^o$, $4^o$, $5^o$, $6^o$, $8^o$, $10^o$, $15^o$, $30^o$,
and $90^o$). The integration time of one individual spectrum was set to 30 s, and each scanning sequence took about 5.5
min. Besides, the controlling electronic devices and connecting fiber are mounted inside. The instrument is equipped
with a high-precision Global Position System (GPS) to record the real-time coordinated positions of the cruise ship.
The detailed description of the setup of MAX-DOAS in CAMS and SUST can be found in Liu et al. (2021).

#### 2.2.2 Data processing and filtering

The MAX-DOAS measurements could be influenced by the exhaust from the measurement ship. Therefore, the data
contaminated by the exhaust were filtered out. As shown in Figure S1(b), the direction and speed of the plume
exhausted from the ship depends on the ship and the true wind speeds/directions. Individual measurements taken
under unfavorable plume directions (plume directions between 45 and 135◦ with respect to the heading of the ship)
were discarded. To avoid the strong influence of the stratospheric absorption, the spectra measured with solar zenith
angle (SZA) lager than 75 °were filtered out. Under these two filtering criteria, 4.9 and 8.3% of all data were rejected
before DOAS analysis (Xing et al., 2017, 2019, 2020).

### 2.2.3 DOAS analysis

The MAX-DOAS measured spectra were analyzed using the software QDOAS which is developed by BIRA-IASB (http://uv-vis aeronomie.be/software/QDOAS/). The DOAS fit results are the differential slant column densities (DSCDs), i.e. the difference of the slant column density (SCD) between the off-zenith and the corresponding zenith reference spectra. Details of the DOAS fit settings are listed in Table1. A typical DOAS retrieval example for the oxygen dimer ($O_4$), nitrogen dioxide ($NO_2$), and nitrous acid (HONO) are shown in Figure 1. The stratospheric contribution was approximately eliminated by taking the zenith spectra of each scan as reference in the DOAS analysis. Before profile retrieval, DOAS fit results of $O_4$, $NO_2$, and HONO with root mean square (RMS) of residuals larger than $3 \times 10^{-3}$ were filtered. Furthermore, the SCD data under the color index (CI) being < 10% of the thresholds obtained through fitting a fifth-order polynomial to CI data which is a function of time was filtered out to ensure a high signal-to-noise ratio (SNR) of the spectra. This filtering criteria remove 2.1, 3.9, and 5.3% for $O_4$, $NO_2$, and HONO, respectively.

### 2.3 Vertical profile retrieval

Aerosol and trace gases (i.e., $NO_2$ and HONO) vertical profiles are retrieved from MAX-DOAS measurements using the algorithm reported by Liu et al. (2021). The inversion algorithm is developed based on the Optical Estimation Method (OEM) (Rodgers, 2000), which employs the radiative transfer model VLIDORT as the forward model. The detailed retrieval procedure is displayed in Appendix I and Figure S3.

In this study, an exponential decreasing a priori with a scale height of 1.0 km was used as the initial profile for both the aerosol and trace gases retrieval (Figure S4). The surface concentrations of aerosol, $NO_2$, and HONO were set to 0.2 $km^{-1}$, 3.0 ppb, and 1.0 ppb, respectively. We assume a fix set of aerosol optical properties with asymmetry parameter of 0.69, a single scattering albedo of 0.90, and ground albedo of 0.05. Furthermore, the uncertainty of the aerosol and trace gases a priori profile was set to 100% and the correlation length was set to 0.5 km. The averaging kernels indicated that the sensitivity of the profile retrieval tended to decrease with increasing altitude, and was especially sensitive to the layers within 0–1.5 km (Figure S5). The sum of the diagonal elements in the averaging kernel matrix is the degrees of freedom (DOF), which denotes the number of independent pieces of information contained in the measurements.

### 2.4 Error analysis

For profile retrieval, the error sources can be divided into four different types: smoothing error, measurement noise error, forward model error, and model parameter error (Rodgers, 2004). However, in terms of this classification, some errors are difficult to be calculated or estimated. For example, forward model error, which is caused by an imperfect representation of the physics of the system, is hard to be quantified due to the difficulty of acquiring an improved forward model. Given calculation convenience and contributing ratios of different errors in total error budget, we mainly took into account error sources based on the following classification, which were smoothing and noise errors, algorithm error, cross section error, and uncertainty related to the aerosol retrieval (only for trace gas). Here, we estimated the contribution of different error sources to the trace gas vertical column densities (VCDs) and AOD, and near-surface (0–200 m) trace gas concentrations and aerosol extinction coefficients (AECs), respectively. The detailed demonstrations and estimation methods are displayed below, and the final results are summarized in Table 3.

a.  Smoothing errors arise from the limited vertical resolution of profile retrieval. Measurement noise errors denote the noise in the spectra (i.e., the fitting error of DOAS fits). They can be quantified by averaging the error of retrieved profiles, as the error of the retrieved state vector equals the sum of these two independent errors. We calculated the sum of smoothing and noise errors on near-surface concentrations and column densities, which were 14 and 5 % for aerosols, 16 and 17 % for $NO_2$, and 20 and 22 % for HONO, respectively in the sea scene. The corresponding values were 13 and 5 % for aerosols, 14 and 16 % for $NO_2$, and 18 and 20 % for HONO, respectively at SUST and 13 and 5 % for aerosols, 15 and 17 % for $NO_2$, and 19 and 21 % for HONO at CAMS.

b.  Algorithm error is the discrepancy between the measured and modelled DSCDs. This error contains forward model error from an imperfect approximation of forward function (e.g., spatial inhomogeneities of absorbers and aerosols), forward model parameter error from selection of parameters, and error not related to the forward function parameters, such as detector noise (Rodgers, 2004). Algorithm error is a function of the viewing angle, and it is difficult to assign this error to each altitude of profile. Usually, the algorithm errors on the near-surface values and column densities are estimated by calculating the average relative differences between the measured and modeled DSCDs at the minimum and maximum elevation angle (except 90°), respectively (Wagner et al., 2004). Considering its trivial role in the total error budget, we estimated these errors on the near-surface values and the column densities at 4 and 8 % for aerosols, 3 and 11 % for $NO_2$, and 20 and 20 % for HONO, according to Wang et al. (2017) and Wang et al. (2020).

c.  Cross section error is the error arising from an uncertainty in the cross section. According to Thalman and Volkamer, (2013), Vandaele et al. (1998), and Stutz et al. (2000), we adopted 4, 3, and 5 % for $O_4$ (aerosols), $NO_2$, and HONO, respectively.

d.  The trace gas profile retrieval error represents the one, which is sourced from aerosol extinction profile retrieval and propagated to retrieved trace gas profile. This error could be roughly estimated based on a linear propagation of the total error budgets of the aerosol retrievals. The errors of trace gases were roughly estimated at 15% for

VCDs and 10% for near-surface concentrations for the two trace gases in the sea scene. The corresponding values
were 14 and 10 % for near-surface concentrations and VCDs, respectively at SUST, and 14 and 10 % at CAMS.
The total uncertainty was calculated by adding all the error terms in the Gaussian error propagation, and the final
results were listed in the bottom row of Table 3. We found that the sum of smoothing and noise errors played a
dominant role in the total uncertainty.

## 2.5 Ancillary data

Meteorological data (including temperature, pressure, relative humidity, visibility, solar radiation intensity, wind
speed, and wind direction) with a temporal resolution of 1 min was measured in the weather station installed on the
ship. NO was measured using NO analyzer (Thermo Scientific model 42i) with a 1 min resolution. The speed of the
ship was calculated referring to the GPS data.
The temperature and relative humidity of two ground-based stations (i.e., CAMS and SUST) were collected from
Weather Underground website, temporal resolution of which is around 3 hours.
The backward trajectory was calculated using HYSPLIT (Hybrid Single-Particle Lagrangian Integrated Trajectory)
developed by the National Oceanic and Atmospheric Administration Air Resource Laboratory (NOAA-ARL). The
meteorological data with a $1^o \times 1^o$ spatial resolution and 24 layers were collected from the Global Data Assimilation
System (GDAS).

## 3 Results and Discussion

### 3.1 Overview of the MAX-DOAS observation over marginal seas of China

A radiative transfer model SCIATRAN was used to convert SCDs of $NO_2$ and HONO to their tropospheric VCDs.
The vertical profiles of aerosol, $NO_2$, and HONO retrieved from MAX-DOAS, the temperature and pressure vertical
profiles simulated using a dynamical-chemical model (WRF-Chem), and the geo-position data collected by GPS were
introduced as inputs in SCIATRAN for the $NO_2$ and HONO air mass factor (AMF) calculation. Missing data are due
to power and instrument system failure, interference of ship plume, unfavorable weather condition (i. e., heavy rain),
and night sailing. During the cruise of Chongming to Zhanjiang, $NO_2$ VCDs varied from $1.05 \times 10^{14}$ to $4.02 \times 10^{16}$
molec.cm$^{-2}$ with an averaged value of $3.90 \times 10^{15}$ molec. cm$^{-2}$. From Zhanjiang to Qingdao, $NO_2$ VCDs varied from
$1.08 \times 10^{14}$ to $2.60 \times 10^{16}$ molec.cm$^{-2}$ with an averaged value of $4.27 \times 10^{15}$ molec. cm$^{-2}$. From Chongming to Zhanjiang,
HONO VCDs varied from $1.00 \times 10^{14}$ to $2.58 \times 10^{15}$ molec. cm$^{-2}$ with a mean value of $2.39 \times 10^{14}$molec. cm$^{-2}$. From
Zhanjiang to Qingdao, HONO VCDs varied from $1.01 \times 10^{14}$ to $2.61 \times 10^{15}$molec. cm$^{-2}$ with a mean value of $2.74 \times 10^{14}$
molec. cm$^{-2}$.
Figure 2 showed the spatial distribution of $NO_2$ and HONO VCDs over the marginal seas of China. Five enhanced
tropospheric $NO_2$ VCDs hot spots were observed during the whole campaign, i.e., the coastal areas of Yangtze River
Delta, Taiwan straits, Guangzhou-Hong Kong-Macao Greater Bay areas, Zhanjiang Port, and Qingdao port. In the
coastal areas of Yangtze River Delta, the hot spots were mainly distributed in the Yangtze River estuary, Hangzhou
Bay, Ningbo port, Taizhou port, and Wenzhou port. These areas are mostly important shipping channels or shipping
ports, and are great $NO_2$ emission sources. The averaged $NO_2$ VCDs in above five areas reached $1.07 \times 10^{16}$, $1.30 \times 10^{16}$,
$7.27 \times 10^{15}$, $5.34 \times 10^{15}$, and $3.12 \times 10^{15}$ molec. cm$^{-2}$, respectively (Figure S6(a)). HONO exhibited similar spatial
distribution characteristics as $NO_2$, and the averaged HONO VCDs in above five hot-spot areas reached $1.01 \times 10^{15}$,
$7.91 \times 10^{14}$, $6.02 \times 10^{14}$, $5.36 \times 10^{14}$, and $5.17 \times 10^{14}$ molec. cm$^{-2}$, respectively (Figure S6(b)). It indicates that $NO_2$ is an
important precursor of HONO. Earlier studies reported that HONO can be generated from $NO_2$ through heterogeneous
reaction on the surface of aerosol and sea (Yang et al., 2021). However, there are obvious differences in the
concentration distribution of HONO and $NO_2$ in the southeast coastal area of Jiangsu (from Qidong to Dongtai). In
this area, $NO_2$ showed a higher concentration ($1.66 \times 10^{16}$ molec. cm$^{-2}$, which is 4 times higher than the mean $NO_2$
VCD), while HONO showed a lower concentration ($2.06 \times 10^{14}$ molec. cm$^{-2}$, which is ~80% of the mean HONO VCD).
It may be the fresh ship emission plume on the route enhancing the $NO_2$ concentration and HONO has not been fully
formed from $NO_2$ heterogeneous reaction in time, since the observations from ship-based MAX-DOAS are
instantaneous.
The surface concentration of $NO_2$ and HONO were extracted from their corresponding vertical profiles. As shown in
Figure 3, the total averaged near-surface $NO_2$ concentrations under sea-oriented and land-oriented measurements were
8.46 and 11.31 ppb, respectively. The total averaged near-surface HONO concentrations were 0.23 and 0.27 ppb
under sea-oriented and land-oriented measurements. The total averaged near-surface HONO/$NO_2$ ratios in
sea-oriented and land-oriented measurements were 0.027 and 0.024, respectively. Earlier studies reported that vehicle
and ship emissions were the main primary HONO sources on land and sea, respectively, and $NO_2$ heterogeneous
reaction on the surfaces of ground, sea, vegetation, and aerosol were the important secondary HONO sources (Liu et
al., 2021). Additionally, they found that the surface HONO concentration under the sea case was lower than that under
the land case, especially in the morning and evening (Yang et al., 2021). Figure 4 showed the time series of AOD, the
surface concentrations of $NO_2$ and HONO, and the surface HONO/$NO_2$ during the whole campaign. We found that
the time series of AOD and $NO_2$ were similar. The high AOD and $NO_2$ usually appeared in busy shipping channels
and ports, and the obvious high-value areas were the coast of the Yangtze River Delta, the Taiwan Strait, Xiamen port,
Zhanjiang port, and Qingdao port (with mean AOD and $NO_2$ of 1.28 and 18.90 ppb, respectively). HONO always
appeared under high AOD and $NO_2$ conditions, however, high AOD and $NO_2$ were not necessarily accompanied with
high HONO concentration. This was because the heterogeneous formation of HONO requires suitable meteorological
conditions (i.e., RH and temperature) in addition to its precursor ($NO_2$) and the reaction surface (aerosol) (Liu et al.,
2019). The high HONO/$NO_2$ values were found on 02, 13, and 14 May with an average value of 0.45. Furthermore,
we found the high values of HONO/$NO_2$ always appeared from 11:00 to 14:00 during a whole day.

## 3.2 Relationship between HONO/$NO_2$ with RH, Temperature, and aerosol in land and sea

Sun et al. (2020) reported that HONO concentrations could increase up to 40–100% over the shipping routes and
international ports, and Huang et al. (2017) reported vehicle exhaust could contribute to ~12–49% of the atmospheric
HONO budget. Since the direct emissions of the measurement ship were removed before data analysis, the primary
source of HONO during the whole campaign was mainly from the direct emissions of cargo ships. By subtracting the
average marine background of $NO_x$ and HONO from the ship plume emission values, the impact of background
values is reduced and the emission ratio of $\Delta HONO/\Delta NOx$ can be obtained, and this emission ratio can be used for
quantifying the primary HONO (Sun et al., 2020; Xu et al., 2015). In this study, we used an averaged 0.46±0.31%
emission ratio of $\Delta HONO/\Delta NOx$ referring to Sun et al. (2020) to understand the primary source of HONO on the sea
surface during the campaign. The NO was measured using in situ instrument, and sea-surface $NO_2$ was extracted from
the retrieved $NO_2$ vertical profiles ($NO_x = NO + NO_2$). Additionally, the calculation method of emission ratios of
$\Delta HONO/\Delta NOx$ in CAMS and SUST was referred from Xu et al. (2015), Liu et al. (2018), and Xing et al. (2021)
(Appendix II). The averaged emission ratios in CAMS and SUST were 0.82±0.34% and 0.79±0.31%, respectively.
The direct emissions were deduced in the following study of the secondary formation of HONO. The ratios of
HONO/$NO_2$ in CAMS, SUST, and the ship-based campaign could be found in Figure S7. Furthermore, the main
secondary formation pathway of HONO is considered as the heterogeneous reaction of $NO_2$ on the surface. The linear
regression between HONO and $NO_2$ in land and static sea scenarios is shown in Figure 5. We found the fitting slopes
in static sea scenes was ~8–10 times larger than that in land scenes, especially on sea-oriented measurements under
static weather condition (slope ≈ 0.06). The correlation coefficients (R) in inland and static sea scenarios were all >
0.62, except in SUST (R = 0.58), which indicates the formation rate of secondary HONO from $NO_2$ heterogeneous
reaction in static sea scenarios may be faster than that in land scenarios. The corresponding temperature and RH
conditions of each spot are displayed in Figure S8, which roughly reveals the impact of RH and temperature on the
process of $NO_2$ forming HONO through heterogeneous reactions.

### 3.2.1 RH dependence on HONO formation

The scatter plots of HONO/$NO_2$ against RH in different land and sea conditions are illustrated in Figure 6. The
highest values can represent varying range of data in each interval and reveal concentration levels of data distribution.
To eliminate the influence of other factors, the average of the six highest HONO/$NO_2$ in each 10% RH interval is
calculated to reflect the distribution range of data in each interval (Liu et al., 2019). The dependence of the averaged
top-6 HONO/$NO_2$ on RH reveal an overall variation tendency of HONO/$NO_2$ against RH. In the inland (CAMS) and
coastal (SUST) cases, the RH turning points are both ~65% (60–70%), where increasing trend switches to decreasing
tendency. The HONO/$NO_2$ increases along with RH when RH is less than 65%, and the HONO/$NO_2$ will decrease
when RH is larger than 65%, which implies that it contributes to the HONO formation from the heterogeneous
reaction of $NO_2$ on wet surfaces with the gradual increase of RH until 65%. The decrease of HONO/$NO_2$ with RH
larger than 65% is presumably due to the efficient uptake of HONO on wet surfaces, and the wet surfaces being less
accessible or less reactive to $NO_2$ when RH being larger than 65% (Liu et al., 2019). However, two turning peaks of
RH were found in the sea cases. The first RH turning peak occurred in ~60%, which is the similar to that under the
inland and coastal cases, and another RH turning peak appeared in ~85% (80–90%). This implies that high RH also
could increase the HONO formation in sea cases. Additionally, the HONO/$NO_2$ decreased sharply when RH was
larger than 95%, because the reaction surface will asymptotically approach a water droplet state to limit the formation
of HONO with RH larger than 95%.

### 3.2.2 Temperature dependence on HONO formation

The scatter plots of HONO/$NO_2$ against temperature in different land and sea conditions are shown in Figure 7.
Similar to the scatter plots of HONO/$NO_2$ against RH, we also adopted the averaged top-6 HONO/$NO_2$ values in each
5℃ interval to represent a general variation tendency of HONO/$NO_2$ against temperature. In the inland condition
(CAMS), the HONO/$NO_2$ decreased along with the increase of temperature, and the highest values of HONO/$NO_2$
appeared at ~12.5℃. However, we found that HONO/$NO_2$ increased along with the increase in temperature, and the
highest values of HONO/$NO_2$ appeared at ~31.5℃ in coastal condition (SUST), which indicates that the HONO
formation from $NO_2$ heterogeneous reaction will be accelerated under lower and higher temperature in the inland and
coastal conditions, respectively. In the sea condition, the HONO/$NO_2$ increased along with the increase in temperature
with a high value under ~25.0℃ when the atmospheric temperature was larger than 18.0℃, and simultaneously, a
~1.9 averaged HONO/$NO_2$ high value was found under ~15.0℃ (14.0–17.0℃). Furthermore, we found that the
appearance of HONO/$NO_2$ high values under lower temperature (14.0–17.0℃) was usually accompanied by land
breeze. Wen et al. (2019) also reported that relatively high temperature could contribute to the formation of HONO in
the sea condition.

### 3.2.3 Impact of aerosol on HONO formation

To further understand the HONO formation from $NO_2$ heterogeneous reaction on aerosol surface, several correlation
analyses were conducted. As shown in Figure 8, the linear regression plot between HONO and aerosol in land and sea
conditions was performed. It was found that the correlation coefficient (R) between HONO and aerosol varied in the
order of coastal (0.55) > sea (0.51) > inland (0.14). Additionally, the fitting slopes under coastal and sea conditions
(0.07) are about 2.3 times larger than that under inland condition (0.03), which implies that the ground surface maybe
more important than aerosol surface during the process of HONO formed from $NO_2$ heterogeneous reaction in the
ground surface layer of the inland. In the coastal and sea conditions, the aerosol and sea are both important in
providing heterogeneous reaction surface for $NO_2$ to form HONO (Cui et al., 2019; Wen et al., 2019; Yang et al.,
2021). Additionally, we found the averaged values of HONO/$NO_2$ were 0.011±0.004, 0.014±0.006, 0.008±0.003, and
0.007±0.003 when aerosol extinctions are 0–0.3, 0.3–0.6, 0.6–0.9 and > 0.9 $km^{-1}$ in the inland case, respectively
(Figure 8(b)). As shown in Figure 8, the high values of HONO/$NO_2$ were mainly under aerosol extinction being less
than 1.0 $km^{-1}$ with averaged values of 0.012±0.006 and 0.090±0.004 in the coastal and sea cases, respectively. It
indicates that aerosol surface plays a more important role in forming HONO through $NO_2$ heterogeneous reaction in
the sea condition than that in the land condition.

### 3.3 Vertical distributions of HONO/$NO_2$ under different aerosol condition in land and sea

To further investigate the height dependence of HONO/$NO_2$ under land and sea conditions, two cases in Pearl River
Delta (PRD) were selected from the whole campaign. As shown in Figure 9, "A" and "B" were under similar aerosol
level (the extinction coefficients in surface layer being 0.45–0.60 $km^{-1}$) and vertical distribution structure, and were all
observed from 10:00 to 11:00 hrs. The instrument viewed sea accompanied with sea wind in "A" named sea scene,
and the instrument viewed land accompanied with land wind in "B" named land scene. The $NO_2$ in the sea and land
scenes have a similar vertical structure, and the $NO_2$ concentration in land scene are larger than that in sea scene
except on the surface layer. The HONO have the same vertical distribution structure in the above two scenes, and the
HONO concentration in the land scene is always larger than that in the sea scene. In Figure 9(e), we found that
HONO/$NO_2$ under 0–400 m in the land scene is higher than that in the sea scene, however, the HONO/$NO_2$ values are
obviously lower in the land scene than that in the sea scene above 400 m. Furthermore, the growth rate of HONO/$NO_2$
with the increase of height in the sea scene is significantly faster than that in the land scene above 400 m. This
indicates the generation rates of HONO sourced from $NO_2$ heterogeneous reaction on aerosol surface in the sea scene
is larger than that in the land scene above 400 m. Under 400 m, the HONO generation rates in the land scene is larger
than that in the sea scene.
Additionally, we selected inland cases (CAMS) to learn the difference of height dependence of HONO/$NO_2$ compared
with sea scenes under different aerosol loads. As shown in Figure 10, the sea and inland scenes had similar aerosol
levels (low aerosol level: < 0.2 $km^{-1}$) and vertical structure. Furthermore, the $NO_2$ and HONO in the sea and inland
scenes have similar vertical structure, although their concentrations in the sea scene are all larger than that in the
inland scene. In Figure 10(d), we found that the HONO/$NO_2$ in the sea scene was obviously larger than that in the
inland scene above 400 m. The HONO/$NO_2$ in the sea scene was about 4.5 times larger than that in the inland scene
especially above 600 m. As shown in Figure 11, the aerosols under the sea and inland scenes exhibited similar
extinction levels (relatively high level: ~0.8 $km^{-1}$) and vertical structure. The $NO_2$ concentration in the sea scene was
higher than that in the inland scene but with a similar vertical structure. The HONO concentration in the sea scene was
lower than that in the inland scene under 400 m, while the concentration in the sea scene was larger than that in the
inland scene above 400 m. In Figure 11 (d), we found the HONO/$NO_2$ in the inland scene was larger than that in the
sea scene under 600 m, while the HONO/$NO_2$ in the sea scene was about 2 times larger than that in the inland scene
above 600 m. All the above cases indicated that the HONO generation rate from $NO_2$ heterogeneous reaction in the
sea scene was larger than that in the inland scene in higher atmospheric layers above 400–600 m. The high-altitude (>
400–600 m) atmospheric parameters in the sea scene were more conductive to promote the HONO formation through
the heterogeneous reaction of $NO_2$. As shown in Figure S9, the ratio of HONO/$NO_2$ also generally increased with the
increase in height above 0.2 km during the whole ship-based campaign. The greatest sensitivity under 1.5 km and the
high degree of freedom (DOF) for aerosol, $NO_2$, and HONO gave confidence in the retrieval results (Figure S10).

### 3.4 Case study

The important factors and precursors to drive the formation of HONO through heterogeneous reaction had complex
evolution and transport characteristics. To further clarify the role of these parameters in the heterogeneous process of
$NO_2$ to form HONO, three typical processes were selected to reveal the favorable conditions for HONO formation at
the sea scene.

### 3.4.1 20 April: A typical transport event

As shown in Figure 12, the aerosol mainly distributed in 0–200 m with a mean extinction coefficient larger than 0.74 km$^{-1}$. NO$_2$ was mainly distributed near the ground surface with a mean concentration of 28.54 ppb before 13:20. The NO$_2$ during this period may come from local ship emissions, as this area is a main shipping channel. From 14:25 to 17:10, a high-concentration NO$_2$ air mass (averaged 13.29 ppb) was found at ~2.0 km. To understand the source of this high-altitude NO$_2$ air mass, we further investigated the possible influence of transport by using the backward trajectories. We calculated 24 h backward trajectories of air masses at 500, 1000, and 2000 m using HYSPLIT (Figure S11). In Figure S11, we found that the dominant wind direction during this period was southeast at all heights, i.e., 500, 1000, and 2000 m. The transport of air masses carried NO$_2$ emitted by ships in Ningbo and Zhoushan ports to main cargo ports of China and Shanghai. Furthermore, the concentration of NO$_2$ was low (averaged 2.32 ppb) near the ground surface from 14:25 to 17:10. As shown in Figure 12 (e) and (g), a low pressure (< 1020 hPa), north dominant wind direction with the wind speed > 12 m/s appeared at the ground surface during this period, which implies that the clean air from north reduced the local surface NO$_2$. The HONO was mainly distributed near the surface with a mean concentration of 0.07 ppb, and the two peaks were found in the early morning (averaged 0.15 ppb) and at 12:15 (averaged 0.11 ppb), respectively (Figure S12). The relatively high concentration of HONO appearing in the early morning was possibly attributed to the accumulation with the stabilization of boundary layer and attenuation of solar radiation after sunset the day before (Xing et al., 2021). The HONO peak appearing at 12:15 may be sourced from the heterogeneous reaction of NO$_2$ on the aerosol surface under a ~80% RH, 18.5℃ temperature, and $1 \times 10^3$ W/m$^2$ SRI conditions.

### 3.4.2 28 April: A typical event of HONO produced from NO$_2$ heterogeneous reaction

From a typical port observation case, the measurement ship moored at Xiamen port on 28 April. As shown in Figure 13, we found two peaks for aerosol and NO$_2$ from 09:00–11:00 and 14:00–16:00, respectively (averaged aerosol extinction coefficient > 0.8 km$^{-1}$, averaged NO$_2$ concentration > 12.0 ppb). NO$_2$ was mainly distributed near the sea surface layer 0–200 m, and a high-concentration NO$_2$ air mass was found from 1.0–2.0 km during 13:00–14:00 due to the short distance transport of NO$_2$ emitted from ships in Xiamen port (Figure S13). However, aerosol appeared in the range of 0.0–2.0 km during 09:00–11:00 and 14:00–16:00. In Figure 13 (g), we found that the wind speeds in the above two peak periods were obviously higher than that in other periods. From 09:00–11:00, the wind speed was ~5.0 m/s with a northwest dominant direction (urban), and the wind speed was ~6.0 m/s with a southeast dominant direction (port gateway) during 14:00–16:00, which indicates that the short-distance high-altitude transport caused the appearance of high-extinction aerosol mass during the above two periods.

Furthermore, we found the high-concentration HONO only appeared at 14:00–16:00 with a 0.57 ppb averaged concentration under 0.9 km, while it was only about 0.14 ppb during 09:00–11:00 period. The slight increase of RH and temperature (Tem) at 14:00–16:00 (RH: ~75.0%, Tem: 23.7℃) may contribute to HONO formation through heterogeneous reaction of NO$_2$ on the aerosol surface than that at 09:00–11:00 (Figure 13 (d)-(e), Section 3.2). Contrarily, the solar radiation intensity (SRI) (~600 W/m$^2$) at 09:00-11:00 was obviously larger than that (~250 W/m$^2$) at 14:00–16:00 (Figure 13 (f)). The higher SRI accelerated the photolysis of HONO during 09:00-11:00 period (Kraus et al., 1998). Therefore, the lower formation rate and higher photolysis rate lead to a significantly lower HONO concentration at 09:00-11:00 than that at 14:00-16:00.

### 3.4.3 03 May: A typical event with unknown HONO source

The measurement ship conducted observation in the sea area near Zhanjiang on 03 May, 2018. As shown in Figure 14, we found that there was an obvious sinking process for aerosol from ~1.0 km during 09:00-16:00, and eventually accumulated near the sea surface with a high extinction coefficient > 0.92 km$^{-1}$. The NO$_2$ was mainly concentrated near the sea surface layer (0–400 m) with an averaged concentration of 8.93 ppb from 08:00 to 09:00. Thereafter, with the rise the planetary boundary layer (PBL) height after sunrise, NO$_2$ was gradually mixed and spread throughout the PBL from 09:00–13:00. During this period, it was accompanied by the increase of the NO$_2$ concentration (averaged 11.2 ppb) under PBL (Figure S14). It is due to the contribution of ship emissions near the sea surface. Contrarily, the regional transport of NO$_2$ from land also increased the NO$_2$ concentration in this area of the sea, with wind speed increasing from 2.5 to 7.8 m/s with a north wind direction from 10:00 to 16:00 (Figure 14 (g)).

Several HONO peaks (> 0.2 ppb) at 0.5–1.0 km were found from 09:45 to 13:00, and the aerosol and NO$_2$ high values were also observed at this height layer, simultaneously, which implies that the heterogeneous reaction of NO$_2$ on aerosol surface is more important than that on the sea surface for HONO source under sea atmosphere. Additionally, HONO concentration obviously elevated after 14:00, especially during 14:00–16:00 (> 0.4 ppb). It may be sourced from heterogeneous reaction of NO$_2$ on the aerosol surface, under RH a being ~92.5% (Figure 14 (d)). The photolysis of HONO also decreased with SRI < 150 W/m$^2$ (Figure 14 (f)) during this period. Furthermore, a HONO peak (> 0.32 ppb) was observed during 16:40–17:10. However, the NO$_2$ concentration always kept low (< 1.5 ppb) after 16:00, and the temperature was lower than 17 ℃ (Figure 14 (e)), which indicates the heterogeneous reaction of NO$_2$ not being the source of the observed HONO peak. The wind was north dominant with an average speed at 7.8 m/s after 15:00,

433 which implies that the regional transport may not be the source of the observed high-concentration of HONO.
434 Furthermore, the SRI was lower than 87.5 W/m$^2$, and it shows the photolysis of nitrate aerosol also not being the
435 source of the elevated HONO. The unknown HONO source in this area of the sea need to be further explored.

## 4 Summary and Conclusions

Currently, many uncertainties in the study of the HONO forming mechanism through the heterogeneous reaction of $NO_2$ exist. Earlier studies mostly focused on the near-surface layer, and the assessment of the contribution of $NO_2$ heterogeneous reaction to HONO formation in the vertical direction of the boundary layer is insufficient. Therefore, we aim to learn the sea-land and vertical differences of the HONO forming mechanism from $NO_2$ heterogeneous reaction and provide deep insights into the distribution characteristics, transforming process, and environmental effects of tropospheric HONO. Ship based MAX-DOAS observations along the marginal seas of China were performed from 19 April to 16 May 2018. Simultaneously, two ground-based MAX-DOAS observations were conducted in the inland station CAMS and the coastal station SUST to measure the aerosol, $NO_2$, and HONO vertical profiles.

Along the cruise route, we found five hot spots with enhanced tropospheric $NO_2$ VCDs in Yangtze River Delta, Taiwan straits, Guangzhou-Hong Kong-Macao Greater Bay areas, Zhanjiang Port, and Qingdao port. Under high-level $NO_2$ conditions in the above five hot spots, we also observed enhanced HONO levels. Contrastingly, the low-concentration HONO accompanied high-level $NO_2$ in the southeast coastline of Jiangsu province. When peak AOD and $NO_2$ conditions were observed, enhanced HONO were observed, although the reverse was not always the case.

To understand the impacts of RH, temperature, and aerosol on the heterogeneous reaction of $NO_2$ to produce HONO, the emission ratios of $\Delta HONO/\Delta NOx$ were calculated to quantify the contribution of the primary HONO source to the total production of HONO. We found that the RH turning points in CAMS and SUST cases were both ~65% (60–70%), whereas two turning peaks (~60% and ~85%) of RH were found in the sea cases. This implied that high RH could contribute to the secondary formation of HONO in sea atmosphere. With increase in temperature, the HONO/$NO_2$ decreased with peak values appearing at ~12.5℃ in CAMS, whereas the HONO/$NO_2$ gradually increased and reached peak values at ~31.5℃ in SUST. In the sea case, when the temperature exceeded 18.0℃, the HONO/$NO_2$ increased with the increasing temperature and achieved peak at ~25.0℃. This indicated that high temperature could promote the secondary formation of HONO in the sea and coastal atmosphere. Additionally, the correlation analysis under different sea-land conditions indicated that the ground surface is more crucial to the formation of HONO from $NO_2$ heterogeneous reaction in the inland case, whereas the aerosol surface contributed more in the coastal and sea cases.

Furthermore, we found that the HONO/$NO_2$ in the sea case was about 4.5 times larger than that in the inland case above 600 m when AEC was ~0.2 km$^{-1}$, and the HONO/$NO_2$ ratio in the sea case was about 2 times larger than that in the inland case above 600 m when AEC was ~0.8 km$^{-1}$, which implied that the generation rate of HONO from $NO_2$ heterogeneous reaction in the sea case is larger than that in the inland case in higher atmospheric layers (> 600 m). To have a deep understanding of three potential contributing factors of HONO production under marine condition, we selected three typical events, which represented the impacts of transport, $NO_2$ heterogeneous reaction, and unknown HONO source, respectively.

**Data availability**

All measurement data used in this study can be made available for scientific purpose upon request to the corresponding author (chliu81@ustc.edu.cn).

**Author contributions**

CX, CL and KL designed the research and organized this paper. CX wrote this paper, and CL and KL edited it. CX, SX, and YS contributed to the retrieval of MAX-DOAS vertical-profile data. CX, YL CZ, QH and SW contributed to data analysis. CX, WT, HW and HL contributed to the MAX-DOAS instrument setup and observations. All above authors contributed to the revision of this manuscript.

**Competing interests**

We declared that none of the authors has any competing interests.

**Acknowledgements**

We would like to also thank Fudan University (Professor Jianmin Chen's group) for organizing the ship-based
campaign and providing meteorological data.

**Financial support**

This research is supported by the National Natural Science Foundation of China (42207113), the Anhui Provincial
Natural Science Foundation (2108085QD180), the Presidential Foundation of the Hefei Institutes of Physical Science,
Chinese Academy Sciences (YZJJ2021QN06), the National Natural Science Foundation of China (41941011,
51778596, 41575021 and 41977184), the Strategic Priority Research Program of the Chinese Academy of Sciences
(XDA23020301), the National High-Resolution Earth Observation Project of China (No. 05-Y30B01-9001-19/20-3).

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

**Table 1. Detailed information of the measurement cruise**

| Cruise NO. | Periods | Measurement cruise |
|---|---|---|
| NO. 1 | 08:50 to 21:02 19 Apr. | Daishan port (30.24$^{o}$N, 122.16$^{o}$E) to Chongming (31.18$^{o}$N, 121.82$^{o}$E) |
| NO. 2 | 05:40 to 17:45 20 Apr. | Sailing around Chongming island |
| NO. 3 | 06:03 21 Apr. to 08:07 03 May | Chongming (31.18$^{o}$N, 121.82$^{o}$E) to Zhanjiang port (21.12$^{o}$N, 110.67$^{o}$E) |
| NO. 4 | 08:07 03 May to 06:52 09 May | Zhanjiang port (21.12$^{o}$N, 110.67$^{o}$E) to Daishan port (30.24$^{o}$N, 122.16$^{o}$E) |
| NO. 5 | 05:40 11 May to 05:55 14 May | Daishan port (30.24$^{o}$N, 122.16$^{o}$E) to Qingdao (35.89$^{o}$N, 120.87$^{o}$E) |
| NO. 6 | 05:55 14 May to 10:00 16 May | Qingdao (35.89$^{o}$N, 120.87$^{o}$E) to Daishan port (30.24$^{o}$N, 122.16$^{o}$E) |


**Table 2. Detailed retrieval settings of $O_4$, $NO_2$, and HONO.**

| Parameter | Data source | Fitting internals (nm) | | |
|---|---|---|---|---|
| | | $O_4$ | $NO_2$ | HONO |
| Wavelength range | | 338-370 | 338-370 | 335-373 |
| $NO_2$ | 298K, $I_0$-corrected, Vandaele et al. (1998) | √ | √ | √ |
| $NO_2$ | 220K, $I_0$-corrected, Vandaele et al. (1998) | √ | √ | √ |
| $O_3$ | 223K, $I_0$-corrected, Serdyuchenko et al. (2014) | √ | √ | √ |
| $O_3$ | 243K, $I_0$-corrected, Serdyuchenko et al. (2014) | √ | √ | √ |
| $O_4$ | 293K, Thalman and Volkamer (2013) | √ | √ | √ |
| HCHO | 298K, Meller and Moortgat (2013) | √ | √ | √ |
| $H_2O$ | HITEMP (Rothman et al. 2010) | × | × | √ |
| BrO | 223K, Fleischmann et al. (2004) | √ | √ | √ |
| HONO | 296K, Stutz et al. (2000) | × | × | √ |
| Ring | Calculated with QDOAS | √ | √ | √ |
| Polynomial degree | | Order 5 | Order 5 | Order 5 |
| Intensity offset | | Constant | Constant | Constant |


* Solar $I_0$ correction; Aliwell et al. (2002).


**Table 3.** Error budget estimation (in %) of the retrieved near-surface (0–200 m) trace gas concentrations and AECs, and trace gas VCDs and AOD.

| | | | Error source | | | | Total |
|---|---|---|---|---|---|---|---|
| | | | Smoothing and noise errors | Algorithm error | Cross section error | Related to the aerosol retrieval (only for trace gases) | |
| Cruise route | Near-surface | aerosol | 14 | 4 | 4 | - | 15 |
| | | $NO_2$ | 16 | 3 | 3 | 15 | 22 |
| | | HONO | 20 | 20 | 5 | 15 | 32 |
| | VCD or AOD | AOD | 5 | 8 | 4 | - | 10 |
| | | $NO_2$ | 17 | 11 | 3 | 10 | 23 |
| | | HONO | 22 | 20 | 5 | 10 | 32 |
| SUST | Near-surface | aerosol | 13 | 4 | 4 | - | 14 |
| | | $NO_2$ | 14 | 3 | 3 | 14 | 20 |
| | | HONO | 18 | 20 | 5 | 14 | 31 |
| | VCD or AOD | AOD | 5 | 8 | 4 | - | 10 |
| | | $NO_2$ | 16 | 11 | 3 | 10 | 22 |
| | | HONO | 20 | 20 | 5 | 10 | 30 |
| CAMS | Near-surface | aerosol | 13 | 4 | 4 | - | 14 |
| | | $NO_2$ | 15 | 3 | 3 | 14 | 21 |
| | | HONO | 19 | 20 | 5 | 14 | 31 |
| | VCD or AOD | AOD | 5 | 8 | 4 | - | 10 |
| | | $NO_2$ | 17 | 11 | 3 | 10 | 23 |
| | | HONO | 21 | 20 | 5 | 10 | 31 |


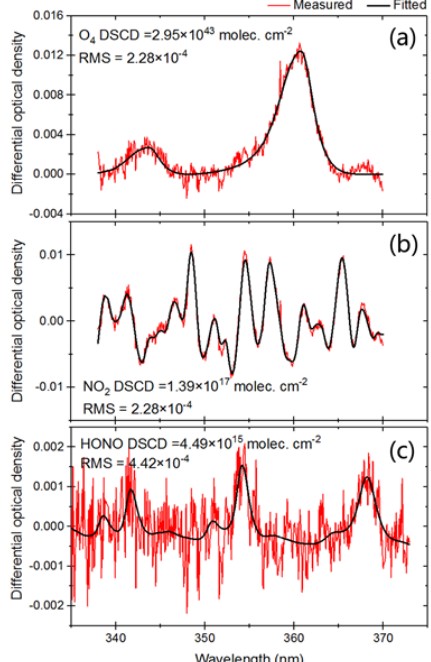

674         Figure 1. Plots depicting typical DOAS spectral fittings for (a) $O_4$, (b) $NO_2$ and (c) HONO.


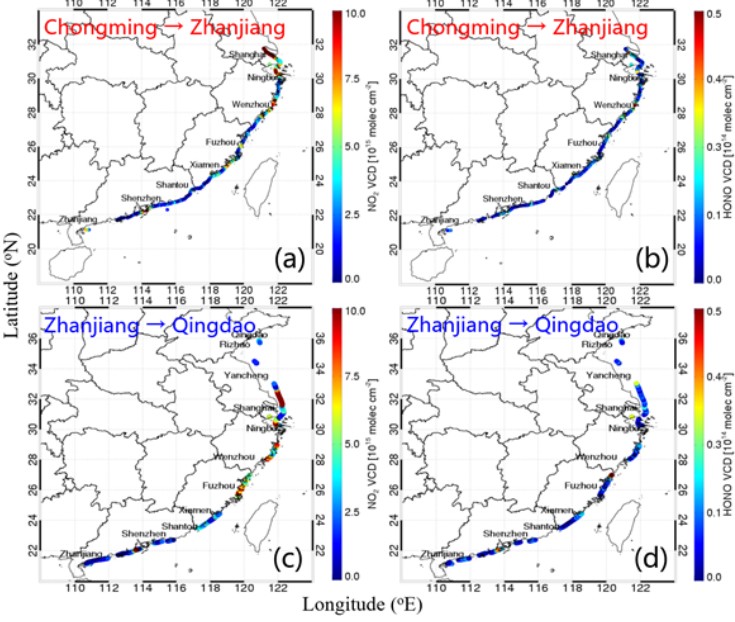

Figure 2. Maps showing the spatial distributions of $NO_2$ and HONO VCDs. (a) and (b) show the $NO_2$ and HONO
VCDs along the cruise route from Chongming to Zhanjiang. (c) and (d) depict the $NO_2$ and HONO VCDs along the
cruise route from Zhanjiang to Qingdao.

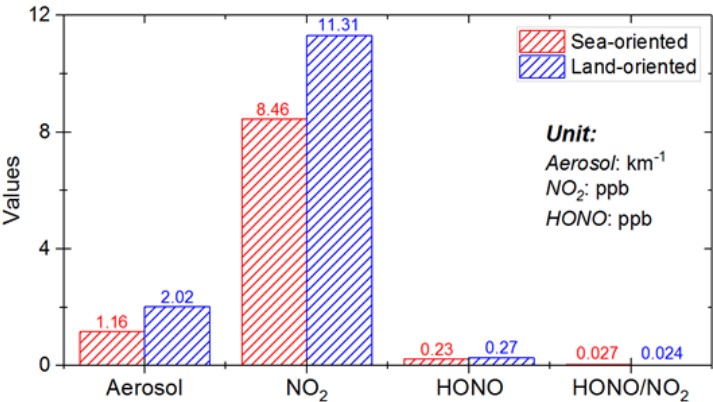

Figure 3. Bar plots of the averaged aerosol extinction, $NO_2$ concentration, HONO concentration, and $HONO/NO_2$ ratio during the campaign. The red and blue boxes denote sea-oriented and land-oriented measurements, respectively.

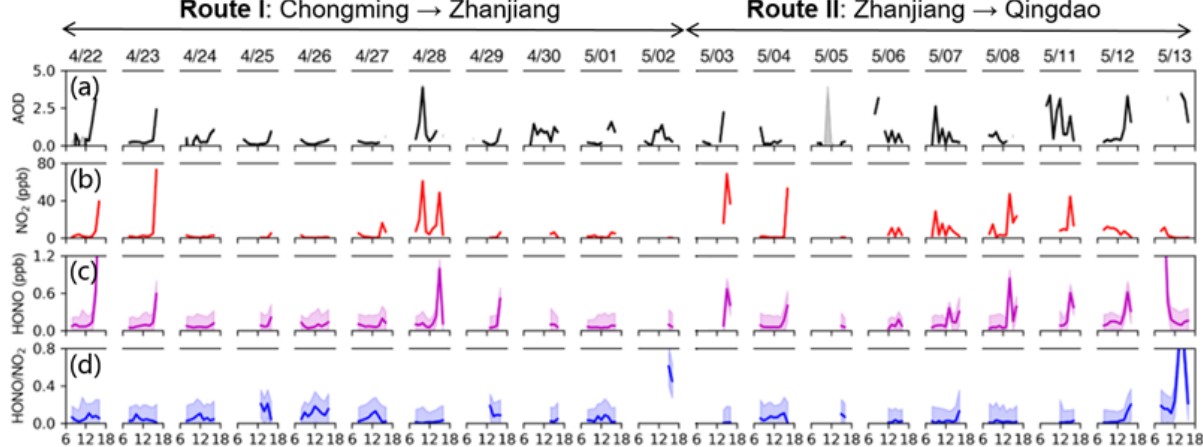

Figure 4. Histograms of the time series of (a) AOD, (b) surface $NO_2$ concentration, (c) surface HONO concentration, and (f) surface $HONO/NO_2$ ratios.

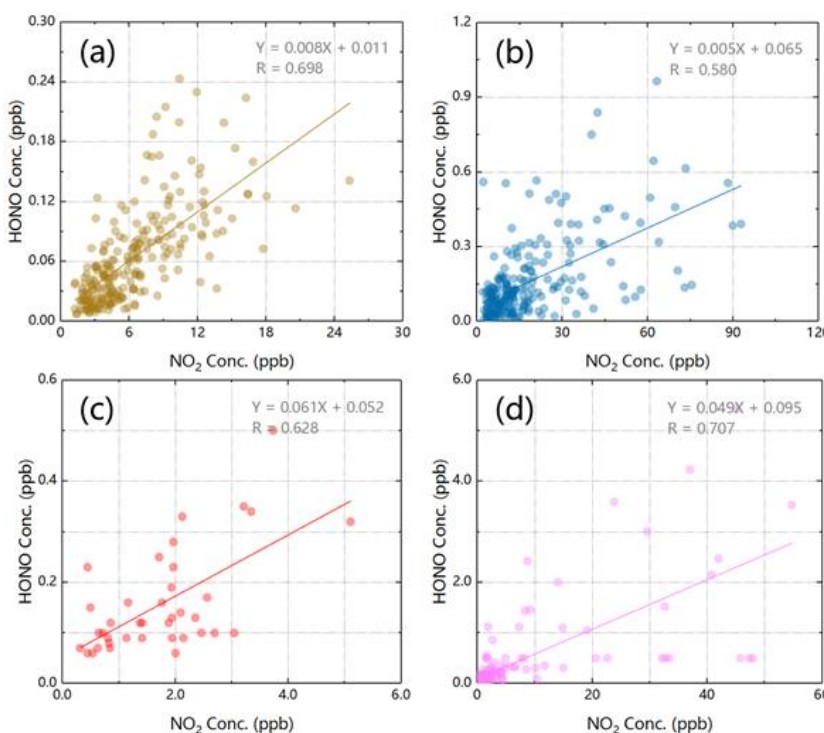

Figure 5. Linear regression plots between surface $NO_2$ and HONO concentrations in (a) CAMS, (b) SUST, and ship-based measurements of (c) sea-oriented and (d) land-oriented under static weather condition.


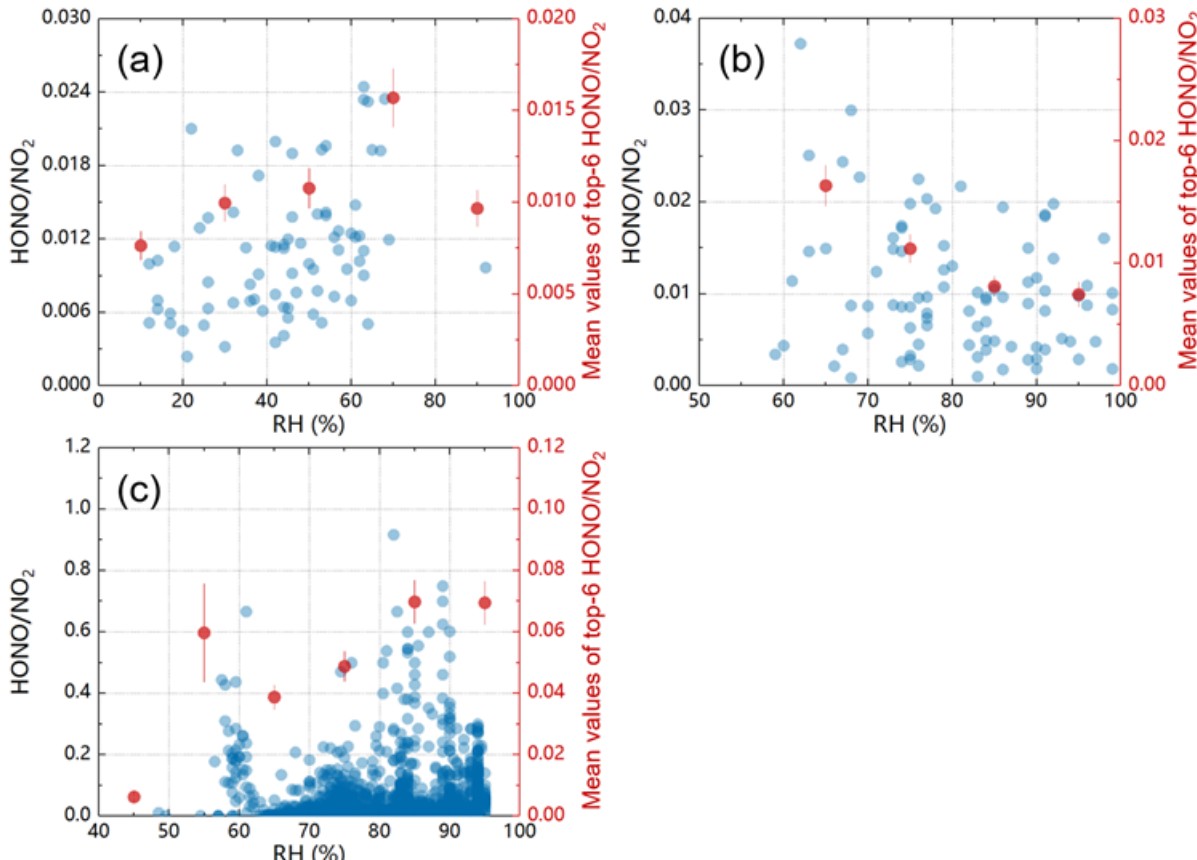

Figure 6. Scatter plots between RH and HONO/NO$_2$ ratios in (a) CAMS, (b) SUST, and (c) the ship-based campaign.

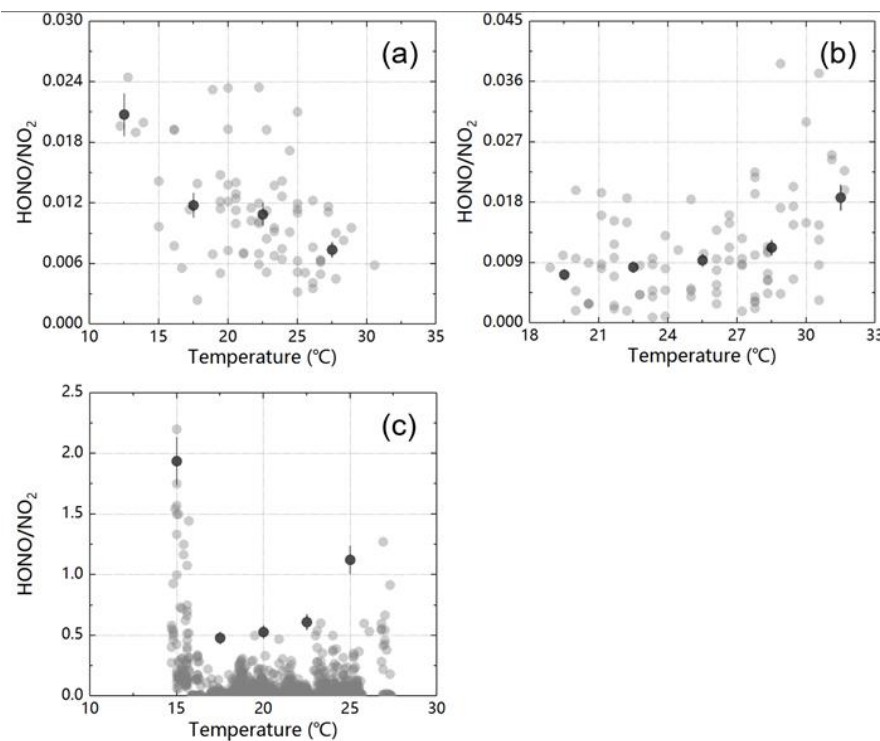

Figure 7. Scatter plots between temperature and HONO/NO$_2$ ratios in (a) CAMS, (b) SUST, and (c) the ship-based
campaign.

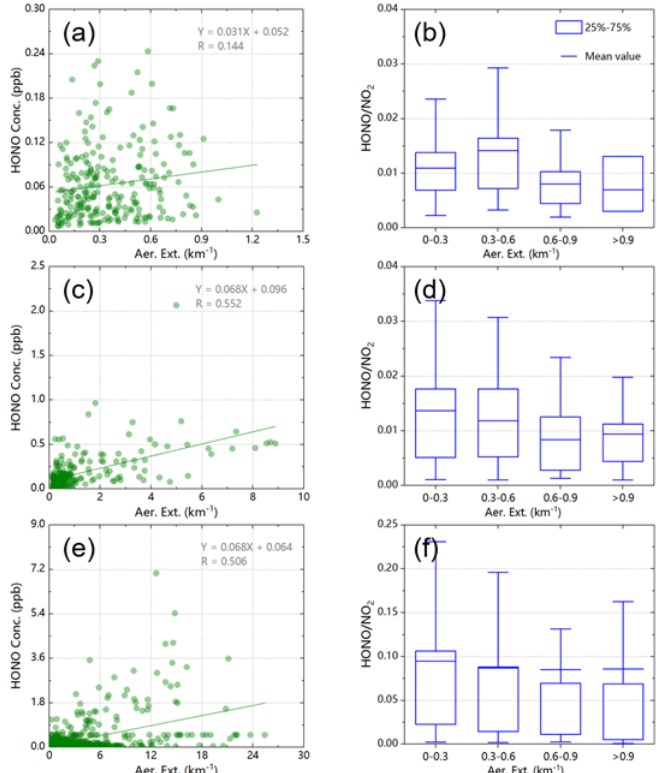

Figure 8. (a), (c), and (e) show the linear regression plots between surface aerosol extinction and HONO
concentrations in CAMS, SUST and the ship-based campaign, respectively. Plots (b), (d), and (f) depicts the
HONO/$NO_2$ ratio distribution under different aerosol extinction coefficient conditions in CAMS, SUST and the
ship-based campaign.

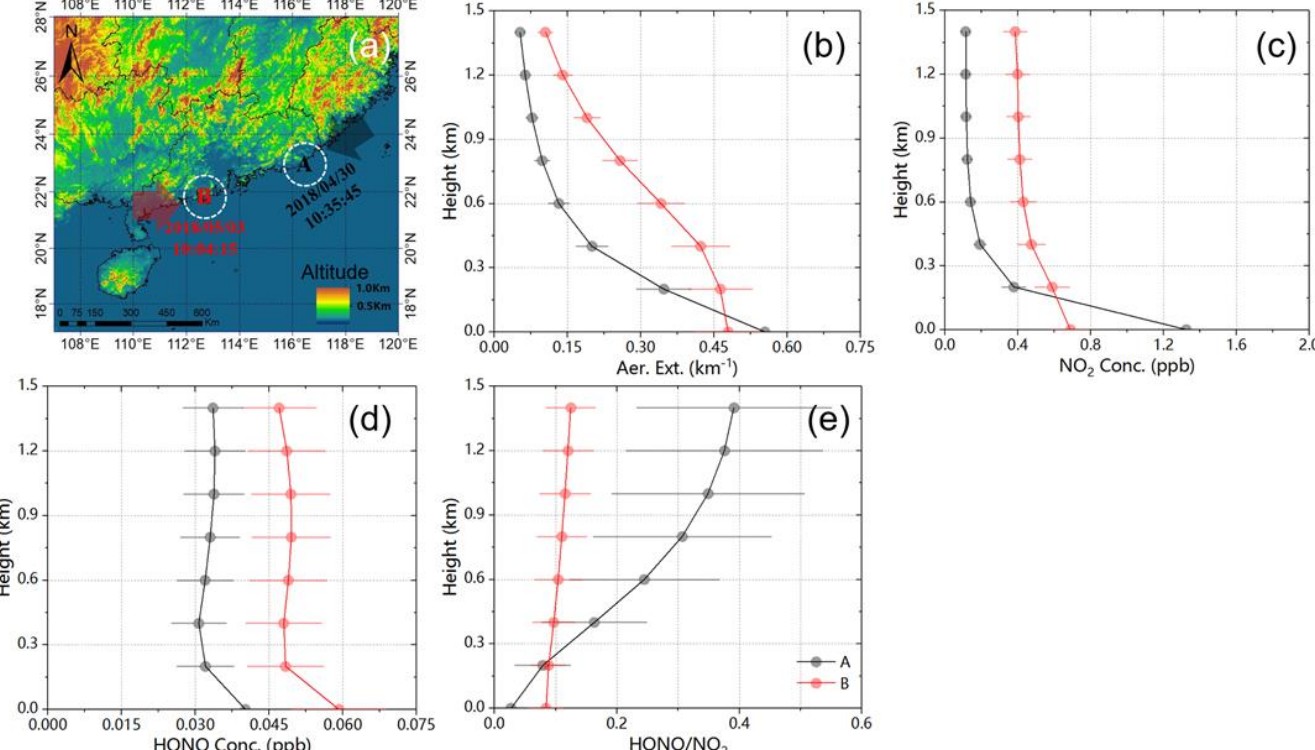

Figure 9. Map (a) shows the two measurement points (A: black, sea-oriented with sea wind; B: red, land-oriented with
land wind) during the campaign. Plots (b)–(e) show the vertical profiles of aerosol, $NO_2$, HONO, and HONO/$NO_2$
ratios in the above two measurement points, respectively.

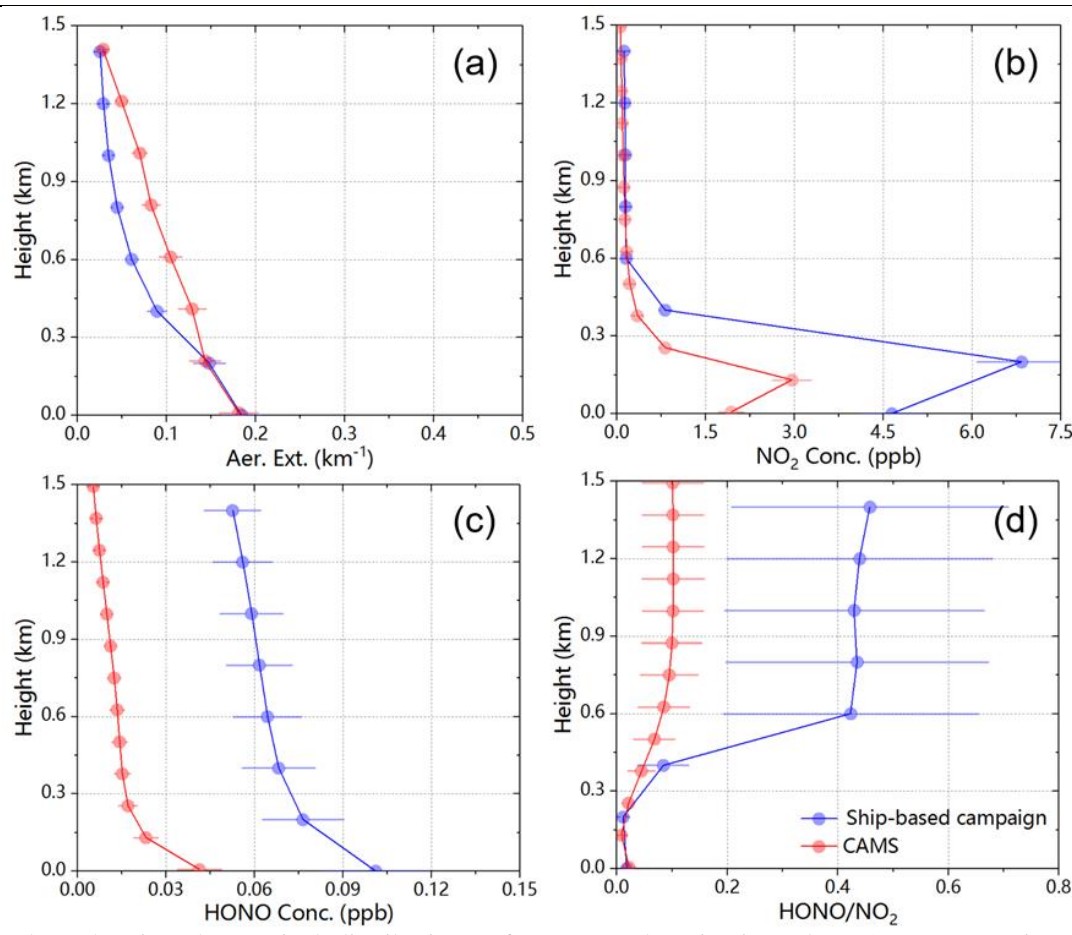

Figure 10. Plots showing the vertical distributions of (a) aerosol extinction, (b) $NO_2$ concentration, (c) HONO
concentration, and (d) HONO/$NO_2$ ratio. The blue and red lines represent a ship-based campaign case and a CAMS
case, respectively.


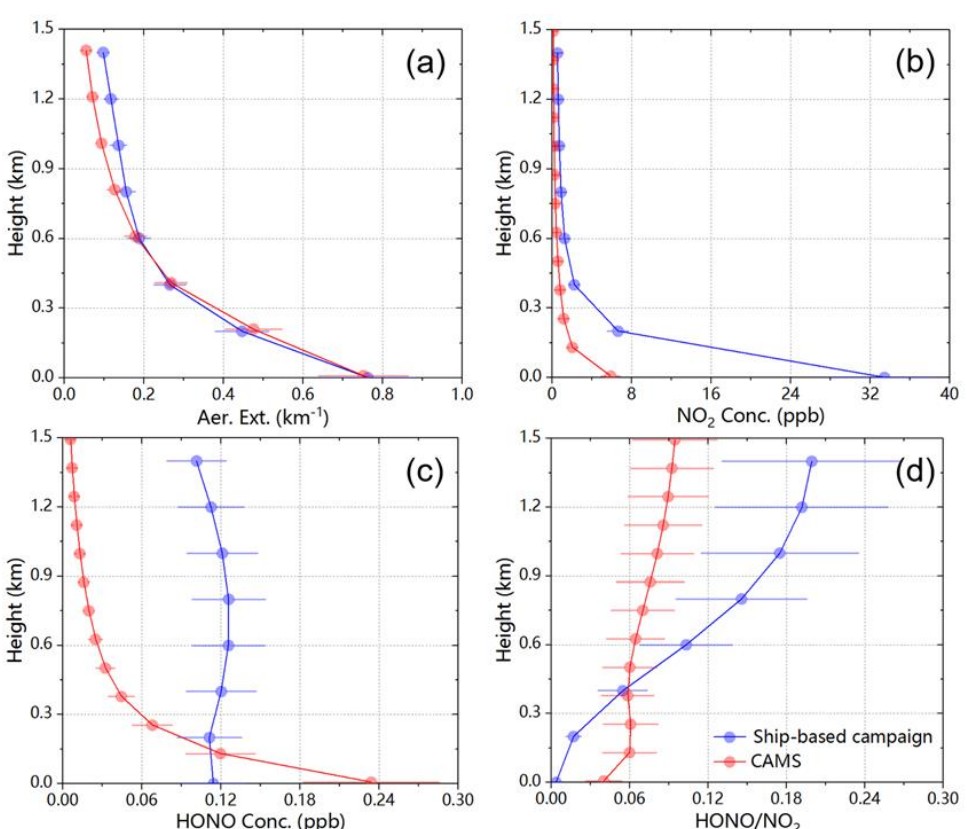


Figure 11. Plots showing the vertical distributions of (a) aerosol extinction, (b) NO$_2$ concentration, (c) HONO
concentration, and (d) HONO/NO$_2$ ratio. The blue and red lines represent a ship-based campaign case and a CAMS
case, respectively.

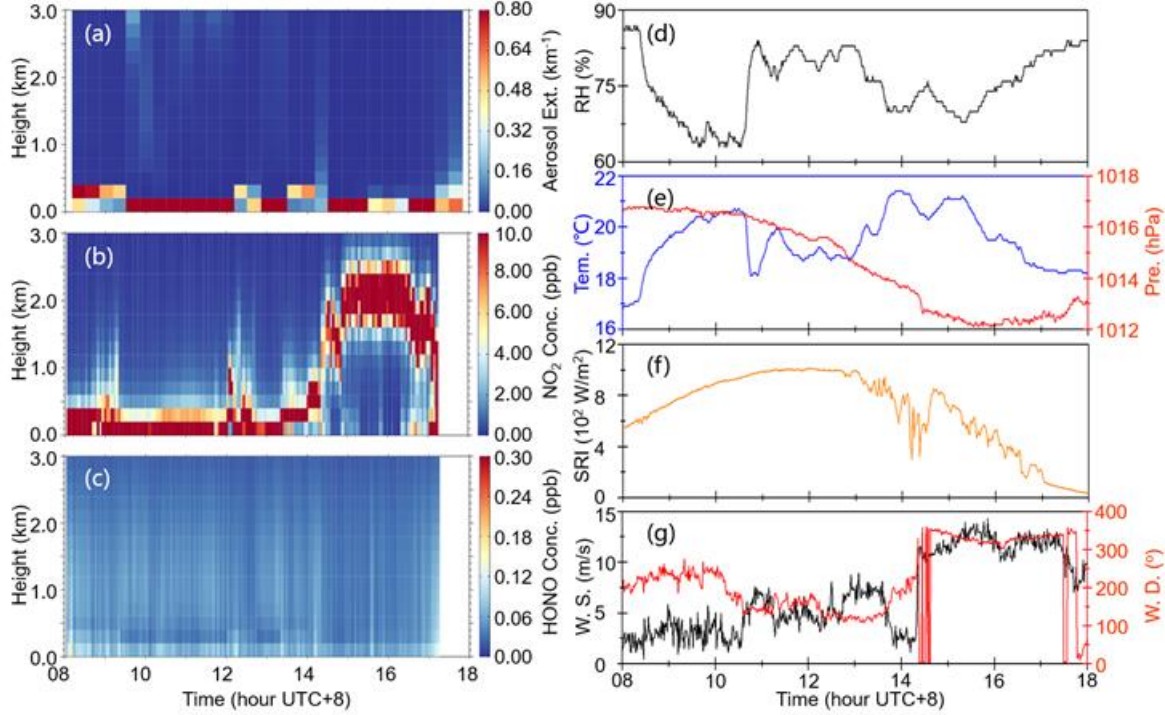

Figure 12. Case of 20 April 2018. Gradient image showing the time series of (a) aerosol extinction, (b) NO$_2$, and (c)
HONO vertical profiles. Plot (d) shows the time series of surface RH. Plot (e) depicts the time series of surface
temperature and pressure. Plot (f) shows the time series of surface SRI. (g) depicted the time series of surface wind
speed and wind direction.


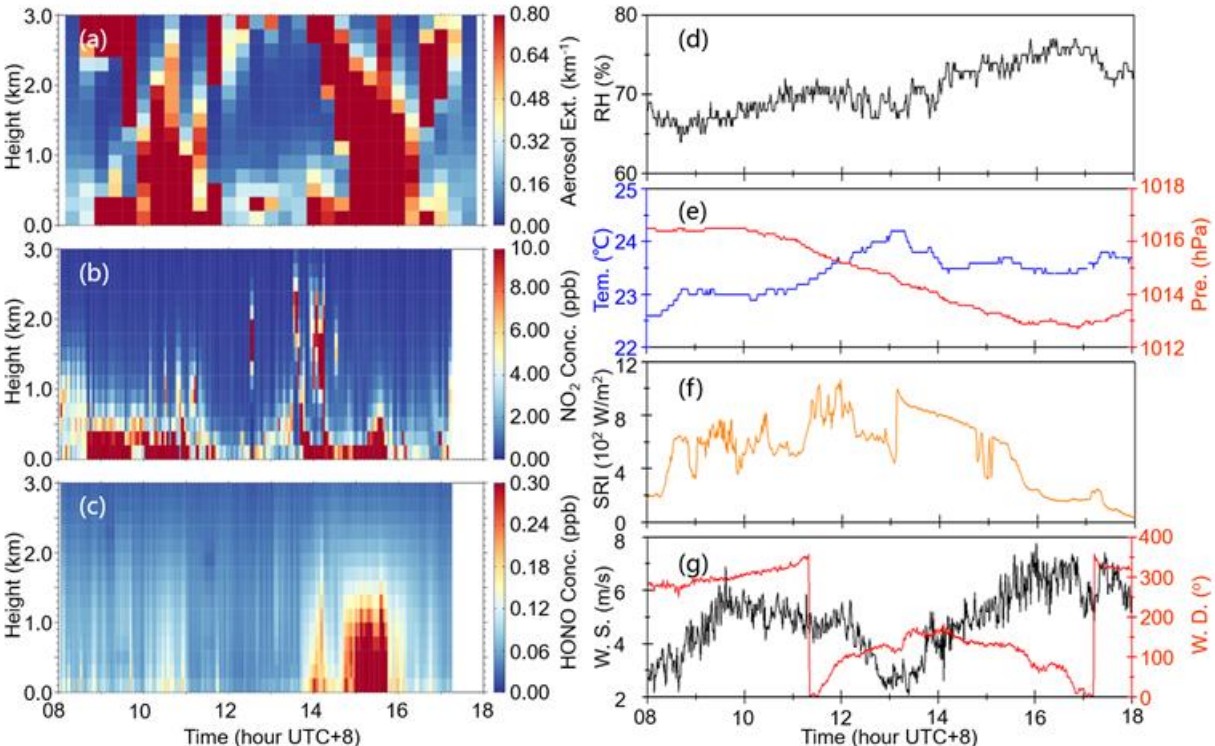


Figure 13. Case of 28 April 2018. Gradient image showing the time series of (a) aerosol extinction, (b) NO₂ and (c)
HONO vertical profiles, respectively. Plot (d) shows the time series of surface RH. Plot (e) depicts the time series of
surface temperature and pressure. Plot (f) shows the time series of surface SRI. Plot (g) depicts the time series of
surface wind speed and wind direction.

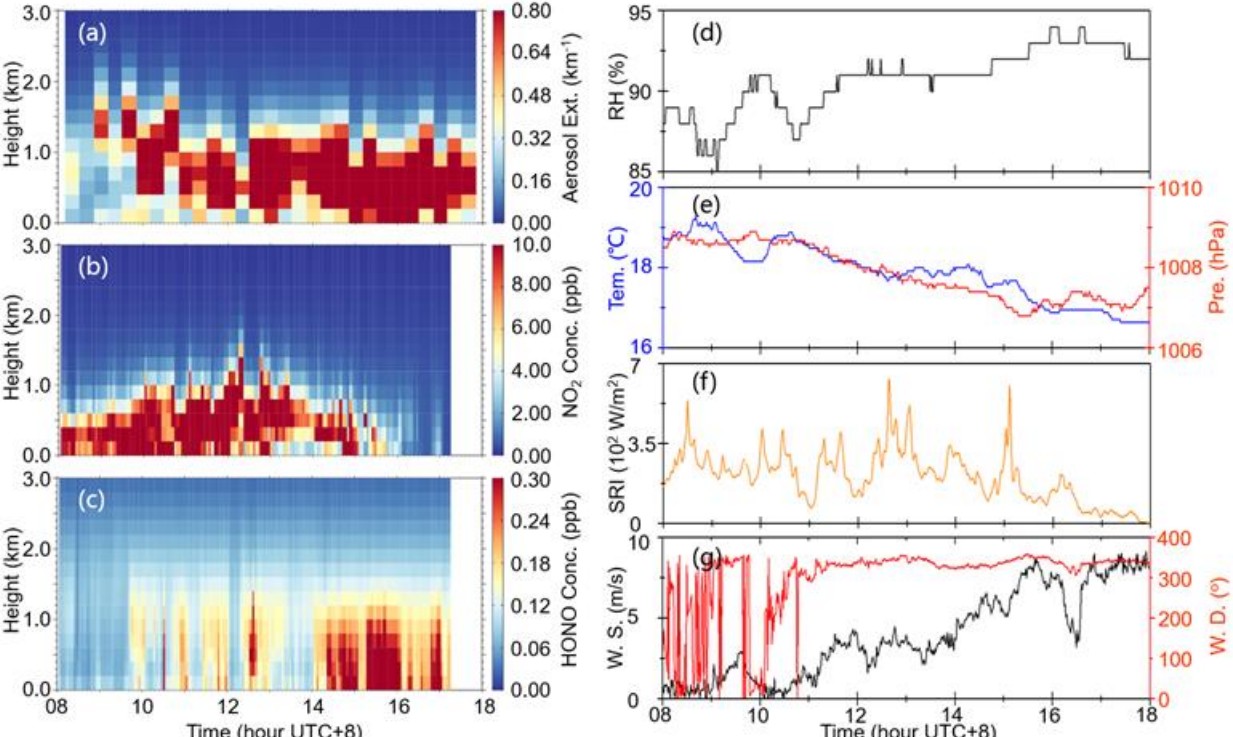

Figure 14. Case of 03 May 2018. Gradient image showing the time series of (a) aerosol extinction, (b) NO₂ and (c)
HONO vertical profiles, respectively. Plot (d) shows the time series of surface RH. Plot (e) depicts the time series of
surface temperature and pressure. Plot (f) shows the time series of surface SRI. Plot (g) depicts the time series of
surface wind speed and wind direction.