# Peer review of "A new insight into the vertical differences in NO2 heterogeneous reaction to produce HONO over inland and marginal seas"

_Atmospheric Chemistry and Physics, 2022_

## Author Comment (AC1)

*Thank you for your careful review and constructive suggestions. These suggestions are quite valuable to us, and help improve our manuscript a lot.*

**Point-to-point responses**

*We appreciate the reviewers for their valuable and constructive comments, which are very helpful for the improvement of the manuscript. We have revised the manuscript carefully according to the Prof. Kleffmann's comments. We have addressed his comments on a point-to-point basis as below for consideration, where the comments are cited in **black**, and the responses are in **blue**.*

In the manuscript by Xing et al. MAX-DOAS measurements during ship cruises and on two land stations (inland and coast) were used to measure vertical gradients of HONO and $NO_2$ to identify potential source mechanisms. Gradient measurements are of significant importance to distinguish between near ground (e.g. direct emissions, heterogeneous $NO_2$ conversion, etc.) and volume sources (e.g. on particles) of HONO. Only when the vertical HONO structure is known, the impact of HONO on the oxidation capacity of the whole boundary layer can be described, in contrast to typical near surface measurements by in-situ instruments, which overweight the contribution of HONO. Also, when using a path averaging spectroscopic method the risk of overestimation of HONO levels by interferences and sampling artefacts in the instrument's inlets are minimized. Thus, such measurements are of general high importance.

However, I could not follow all the evaluations and arguments in the manuscript caused by missing information. The following comments could be considered to improve the manuscript.

**Major comments:**
1) Section 2.1: Missing information to CAMS and SUST sites:
Besides the ship measurements, MAX-DOAS measurements were also performed in parallel in two stations, which were defined as "inland" (CAMS) and "coastal" (SUST). Here I am missing more information to both sites. Especially, where are they? E.g. for the Chinese Academy of Meteorological Science (CAMS) I found Beijing (?), which would be far away from the ship measurements and would make any comparison highly uncertain…
Re: Thanks for your great comments.
As shown in Figure R1, Chinese Academy of Meteorological Sciences (CAMS) was located in the urban of Beijing (116.32ºE, 39.94ºN), and South University of Science and Technology (SUST) was located in Shenzhen (114.00ºE, 22.60ºN). These two MAX-DOAS stations were selected as inland and coastal cases to further understand the impacts of relative humidity (RH), temperature, and solar radiation intensity (SRI) on the heterogeneous reaction of $NO_2$ to form HONO in different scenes.
In order to illustrate the representativeness of CAMS as an inland scene, we selected another MAX-DOAS station (HNU: Huaibei Normal University) shown in Figure R1. Considering that there was no observation data at this station in 2018, we selected HONO and $NO_2$ data at the same time as ship based MAX-DOAS measurements in 2019. Moreover, we only analyzed the vertical distribution of HONO/$NO_2$ in HNU, due to its lack of meteorological data. As shown in Figure R2, we could find that

HONO/NO₂ decreased with the increase of height, which was the same as its performance in CAMS. Wang et al. (2020) and Meng et al. (2022) also reported the conclusion that HONO/NO₂ decreased with the increase of height. Therefore, CAMS station can represent inland scene to some extent, although it was far from the cruise route.

[Figure]

Figure R1. Cruise route and MAX-DOAS stations (CAMS, HNU and SUST).

[Figure]

Figure R2. Vertical profiles of (a) HONO, (b) NO₂, and (c) HONO/NO₂ at HNU station from 19 April to 16 May 2019.

2) Sea- vs. land-oriented measurements:

The ship data was divided in sea- and land-oriented measurements. But isn't that both sea data? To answer this question, two important information are missing: a) How far away were the ship tracks on average from the coastline? b) what is the typical distance for the light-path of the MAX-DOAS (only the horizontal vector is of importance)? I expect that the distance of the ship from the coast (some km?) was larger than the "horizontal view" of the instrument (horizontal distance between the average scattering point and the instrument). From my experience for Chinese conditions the visibility if often significantly smaller than 1 km… In this case the instrument is only evaluating sea influenced air masses and the observed differences

reflect only some undefined horizontal gradient between sea and land, but not any "sea" of "land" data.

Re: Thanks for your great comments.

The average distance between ship and coastline was 2-20 km during the observation. The effective optical path $L$ was calculated using following equation:

$$L = SCD_{O_4} / C_{O_4}$$

Where, $SCD_{O_4}$ was the slant column density of O$_4$, $C_{O_4}$ was the concentration of O$_4$.

The average $L$ of this observation was 2-5 km. Indeed, $L$ was less than the distance between ship and coastline. Moreover, we only selected data observed during clear days with visibility > 10.0 km.

Sea-oriented and land-oriented measurements can reflect the air masses affected by sea and land to some extent, respectively. Figure 3 also reported that the concentrations of aerosol, NO$_2$ and HONO in land-oriented measurements were all larger than that in sea-oriented measurements, considering their more obvious land sources.

In the manuscript, we modified the expression using "land-oriented measurements" and "sea-oriented measurements".

3) Direct HONO/NO$_x$ emission ratio

In section 3.2. it seems that HONO/NO$_x$ ratios from direct emission were determined by the measurement data for CAMS and SUST. However, it is unclear how this has been done? In the present study, only daytime data could be used (light source of the MAX-DOAS = sun…). But one filter to determine the HONO/NO$_x$ ratio of direct emissions from field data - besides others - is to use only night-time data, caused by the fast photolysis of HONO!? In addition, because of strong vertical gradients and the vertical resolution of the MAX-DOAS the combined use of path averaged HONO and NO$_2$ data in comparison to in-situ NO ground data cannot be recommended (apples and oranges…). The method used is completely unclear and should be further explained. E.g. how was the direct emission ratio of 0.46% (line 216) of Sun et al. considered ("used to understand…")?

Re: Thanks for your great comments.

The MAX-DOAS measurements could be influenced by the exhaust from the measurement ship. Therefore, the data contaminated by the exhaust were filtered out. As shown in Figure R3, the direction and speed of the plume exhausted from the ship depends on the ship direction/speed and the true wind speed/direction. Individual measurements taken under unfavorable plume directions (plume directions between 45 and 135∘ with respect to the heading of the ship) were discarded. *HONO/NO$_x$* ratios from direct emission were determined by the measurement data for CAMS and SUST. We derived the emitted *HONO/NO$_x$* ratio referring to the reports in Xu et al. (2015), Liu et al. (2018) and Xing et al. (2021). The fresh plumes were selected using the following criteria: (a) [$NO_x$]>40 ppb, (b) $NO/NO_x$>0.85, (c) good correlation performing between *HONO* and $NO_x$ (R>0.90), (d) short duration of plumes (<=2.0 h), and (e) 70°<SZA<75°. We put above criteria in the revised supplyment.

MAX-DOAS performed based on the collected solar scattering spectrum to retrieve aerosol, *NO$_2$* and *HONO*. In general, we believed that the retrieved MAX-DOAS data was reliable, when SZA was not large than 75°. We usually selected data with 70°<SZA<75° to calculated *HONO/NO$_x$* ratios from direct emission. In this condition, the photolysis rate of *NO$_2$* was not large than $0.25 \times 10^{-3}$ s$^{-1}$.

Surface $NO_2$ was extracted from the retrieved $NO_2$ vertical profiles. As shown in Figure R4, the correlation coefficient (R) between surface $NO_2$ retrieved from MAX-DOAS and in situ $NO_2$ in five stations was large than 0.7 (Song et al., 2022). Therefore, we think that 0-100 m $NO_2$ retrieved from MAX-DOAS measurements can characterize ground surface $NO_2$. Moreover, Ryan et al. (2018) also used data retrieved from MAX-DOAS successfully revealed the $HONO/NO_x$ ratios from direct emission. The key problem here was how to improve the data accuracy of MAX-DOAS in the future.

For Sun et al. (2020), the detailed selection criteria of ship plumes include (a) only the data when the vessel stopped and the plume moved through the optical path were considered; and (b) concentration spikes of $HONO$ and $NO_x$ as well as reduction in $O_3$ concentrations were observed.

[Figure]

Figure R3. (a) Illustration of the MAX-DOAS setup location on the measurement ship. The red rectangle indicates the ship's exhaust. The blue rectangle represents the MAX-DOAS instrument. The blue rectangle represents the meteorological station. (b) The apparent speed and direction of plume.

[Figure]

Figure R4. (a) Correlation analysis of in situ measured PM$_{2.5}$ and surface AECs (0–100 m) retrieved from CAMS, HNU, NC, and SJZ MAX-DOAS stations from January to March, 2021 and (b) their corresponding $NO_2$ comparative results. The black line denotes the linear least-squares fit to the data; $R$ denotes Pearson correlation coefficient; $N$ denotes the number of valid data. (Song et al., 2022)

4) Unrealistic HONO/NO$_x$ data:

If the HONO/NOx ratio for direct emissions of 0.82 % (CAMS) and 0.79 % (SUST) are true, then the slopes of all HONO against $NO_2$ data shown in Fig. 5 (a) 0.8 % for CAMS and b) 0.5 % for SUST) are not possible. Even if one assumes the absence of any NO in the atmosphere (very unreasonable) the slopes when using all data should

be by definition larger than only the direct emission ratio!? Typically, that should be a few % for field data (cf. ratio of the average ship data of ca. 2.5 %, which I get from the data in lines 191-192) for which 0.8 % (lower limit during daytime, see below) may be direct emissions. But here for SUST all data show a lower $HONO/NO_2$ ratio (and the $HONO/NO_x$ ratio would be even much lower…) than the direct emission ratio. Please check the data.

In addition, during daytime a measured $HONO/NO_x$ ratio (e.g. from sharp plumes) will be lower than what is directly emitted. This can be explained by the different lifetimes of HONO (10-20 min during daytime) and $NO_2$ (typically some hours). Thus, depending on the time between emission and measurements the contribution of direct emitted HONO will decrease (this is the reason why the "night-time filter" is used to measure direct emission from field data…). For details I recommend the paper by Xue et al. (https://doi.org/10.5194/acp-22-3149-2022).

Re: Thanks for your great comments.

As shown in Figure R5, we calculated the *HONO/NO₂* ratios in CAMS, SUST and the cruise during the observation. The average *HONO/NO₂* ratios in CAMS and SUST were 0.012 and 0.014, respectively, which were significantly higher than corresponding fitting slopes and the *HONO/NO₂* emission ratios. The average *HONO/NO₂* ratios during the cruise were 0.20-0.25. We put this figure in the revised supplyment.

[Figure]

Figure R5. HONO/NO₂ ratios in CAMS, SUST and the cruise.

The fresh plumes were selected using the following criteria: (a) [*NOₓ*]>40 ppb, (b) *NO/NOₓ*>0.85, (c) good correlation performing between *HONO* and *NOₓ* (R>0.90), (d) short duration of plumes (<=2.0 h), and (e) 70°<SZA<75°. As we all know, MAX-DOAS performed based on the collected solar scattering spectrum to retrieve aerosol, *NO₂* and *HONO*. In general, we believed that the retrieved MAX-DOAS data was reliable, when SZA was not large than 75°. In order to reduce the influence of fast photolysis of *HONO* and *NO₂*, we usually selected data with 70°<SZA<75° to calculated *HONO/NOₓ* ratios from direct emission. In this condition, the photolysis rate of *NO₂* was not large than $0.25 \times 10^{-3}$ s$^{-1}$. We also have learned the paper of Xue et al. (2022).

5) Unrealistic HONO/NO₂ gradient data:

In figures 9-11 vertical gradient data of the HONO/NO₂ ratio are shown. Here increasing ratios are observed with altitude, which is in contrast to most gradient data,

which I know (cf. e.g. our gradient data on a 190 m tall tower, Kleffmann et al., 2003 doi: 10.1016/S1352-2310(03)00242-5). While this may be explained by any unusual chemistry over sea surfaces, the absolute numbers of the HONO/NO$_2$ at higher altitude of up to 45 % (see Fig. 10) are impossible, independent of how strong any HONO source – e.g. particle nitrate photolysis – may be. The photolysis of HONO is a source of NO. In a typical atmosphere for which [O$_3$]>[NO$_x$] this is quickly converted to NO$_2$. Since in higher layers in a well-mixed atmosphere a PSS can be assumed (far away from any direct sources) the maximum HONO/NO$_2$ ratio is given by the ratio of the lifetimes of both molecules. For HONO this is around 10 min at noon (check for J(HONO)), while for NO$_2$ this is mainly limited by its reaction with the OH radical during daytime (the Leighton chemistry will not play a role here). Assuming a high OH concentration of 10$^7$ cm$^{-3}$ at 1 km altitude a lifetime of ca. 3 h can be calculated. Thus, a maximum HONO/NO$_2$ ratio of ca. 6 % should result under steady state conditions. If HONO is measured close to a source, e.g. in near ground measurements in a step vertical gradient, higher HONO/NO$_2$ ratios are possible (= no PSS…). But in a homogeneous mixed atmosphere at 1 km altitude (see figures 9-11) such high HONO/NO2 data is impossible. Please check.

Re: Thanks for your great comments.

As shown in Figure 10, the *HONO/NO$_2$* ratio in CAMS was decreasing with the increase of height under 200 m with aerosol extinction coefficient less than 0.2 km$^{-1}$. The average *HONO/NO$_2$* in CAMS under 200 m was 0.015 during the campaign, which was within the range of *HONO/NO$_2$* (0.0-0.07) in previous studies (Kleffmann et al., 2003; Meng et al., 2020). Moreover, Zhang et al. (2020) also reported that the *HONO/NO$_2$* ratio in Beijing increased with the increase of height under 200 m in haze days. Figure 11 also told us that the *HONO/NO$_2$* ratio in CAMS also increased with the increase of height under the condition of extinction coefficient larger than 0.7 km$^{-1}$.

In order to understand the accuracy of MAX-DOAS data, we analyzed the retrieval quality of MAX-DOAS data described in Figure 10-11 as following.

[Figure]

Figure R6. The top row presented the vertical profiles and errors of aerosol, $NO_2$ and $HONO$ under low aerosol and high aerosol conditions. The bottom row showed the corresponding retrieved averaging kernels.

Figure R6 told us that the data quality was reliable. This section was put into the revised supplyment.

*About the high HONO/NO₂ ratio (~0.45) during the cruise observation (Figure 10):*

We could find that there was an obvious mutation in $HONO/NO_2$ ratio at about 0.5 km. The HONO air mass above 0.5 km maybe detected during this process. As shown in Figure R7, we plotted all the $HONO/NO_2$ ratios during the cruise observation. We also could find the increase of $HONO/NO_2$ with the increase of height. This figure was put into the revised supplyment.

[Figure]

Figure R7. Vertical profiles of (a) aerosol extinction, (b) NO₂, (c) HONO, and HONO/NO₂ ratios during the cruise observation.

**Minor comments in the order of the manuscript:**

Line 37-38: There are several "heterogeneous reactions of NO₂". Here the authors should distinguish between slower nighttime conversion (NO₂+H₂O and NO₂+organic) and daytime sources (NO₂+organic + light, see Stemmler et al., 2006; or NO₂ +TiO₂+light = photocatalysis). Otherwise some arguments of the authors (with solar radiation, see below) are unclear.

Re: Thanks for your great comments.

We have rewritten this sentence as following:

"the known sources of HONO mainly include direct emissions from vehicles, ships, biomass burning and soil, the homogeneous reaction of NO and OH radicals, the nighttime and daytime heterogeneous reaction of NO₂ on aerosols, vegetation, ground and other types of surfaces, and the photolysis of nitrate particles ($NO_3^-$) (Stemmer et al., 2006; Indarto et al., 2012; Wang et al., 2015)."

Line 51-53, general comment to this section, but also to the author's own evaluations: These simple correlation studies always bear the risk of a misinterpretation of the results. Typically, trace gases which are emitted or formed near to the ground will anyhow correlate caused by the variable mixing layer height. The is mainly modulated by diurnal surface temperature variation which has also an effect on the relative humidity. Thus, e.g. at the end of the night the temperature and mixing height are low, while the relative humidity is high. Caused by the resulting high S/V ratio under these conditions, heterogeneous HONO formation is faster and the HONO/NO$_x$ ratio will correlate with the humidity, without any necessary mechanistic link (see also correlation of Radon with HONO…). Also, often at very high humidity the HONO/NO$_x$ ratio is again decreasing with humidity. This is typically explained by

uptake on very humid surfaces. However, the highest relative humidity is often observed close before sunrise, when direct emissions start to increase. Thus, the high HONO/NO$_x$ air masses from slow nighttime sources (typically 5 %) are "diluted" by fresh low HONO/NO$_x$ emissions (around 1%), leading to the decreasing HONO/NO$_x$ ratios at high humidity. Thus, the authors should highlight (and later consider for their own evaluation…) that simple correlation analysis may lead to artificial correlations and misleading conclusions.

Re: Thanks for your great comments. This suggested that more detailed process analysis and quantitative analysis in addition to linear regression analysis should be valued in the future. In this process, with the help of multiple models and cooperation with superior teams, data advantages can be better played.

*We have rewritten these sentences as following:*

Previous works always used the linear regression relationship between HONO/NO$_2$ and above parameters to characterize the influence of these parameters on the formation of HONO through the heterogeneous reaction of NO$_2$. Although this kind of simple linear regression method may lead to artificial correlations and misleading conclusions, considering the vertical evolution of atmospheric parameters. Wen et al. (2019) found that the increased temperature could promote the heterogeneous reaction of NO$_2$ to form HONO in sea conditions. The generation rate of HONO could increase rapidly, when the temperature is greater than 20 ℃. Gil et al. (2019) found that the HONO formed from the heterogeneous reaction of NO$_2$ will increase along with the increase of RH when RH is less than 80% in a case of land park using deep learning forced by measurement results. Fu et al. (2019) reported that RH and SRI are the main parameters driving the heterogeneous reaction of NO$_2$ to form HONO in Pearl River Delta, and it contributes 72% of the total source of HONO. Cui et al. (2019) found that the potential of heterogeneous reaction of NO$_2$ to form HONO will increase with the increase of particle concentration and the specific surface area of single particle in coastal cities.

Line 77-85: With respect to the main topic of the manuscript, I would expect a more extended summary of the existing gradient data (from towers, and MAX-DOAS), which is normally very different to the present results (see major comment 6).

Re: Thanks for your great comments.

Taking tower and aircraft as platforms, these techniques performed to measure HONO vertical profiles, and found that the peak values of HONO usually appeared under 200 m at urban and suburban areas (Kleffmann et al., 2003; Stemmler et al., 2006; Zhang et al., 2009; Wong et al., 2012; Meng et al., 2020; Zhang et al., 2020). These studies also revealed that the heterogeneous reaction of NO$_2$ on multiple surfaces (ground and aerosol etc.) was an important source of HONO under planetary boundary layer (PBL), especially in haze days. Moreover, they also reported that the HONO/NO$_2$ ratios usually decreased with the increase of height under 200m at inland and coastal areas. However, the cost of above techniques used to measure HONO vertical profiles was too high, and the real-time and continuous measurement cannot be realized. Multi-axis differential optical absorption spectroscopy (MAX-DOAS), as a ground-based ultra-hyperspectral remote sensing technology, has been widely used for vertical observation of atmospheric pollutants in the past two decades. In the past five years, several researchers have carried out campaigns based on MAX-DOAS to measure the vertical profile of HONO in inland and coastal areas, and revealed their vertical characteristics, sources and the contribution to atmospheric oxidation at different height layers (Garcia-Nieto et al., 2018; Ryan et al., 2018; Wang et al., 2020;

Xing et al., 2021; Xu et al., 2021; He et al., 2023). There were few studies on the sources of HONO at different height layers in sea conditions. In this study, it will be the first time to use MAX-DOAS to study the spatiotemporal distribution and the sources of HONO along the Chinese coastline, and to learn the differences of the HONO formed from the heterogeneous reaction of $NO_2$ in different height layers and land-sea scenes.

Line 187-189: This sentence could make sense only if a photolytic $NO_2$ conversion process is considered (see above). However, even for a photolytic $NO_2$ conversion process which was found to correlate with $J(NO_2)$ in lab studies (see Stemmler et al., 2006), the steady state $HONO/NO_2$ ratio would not change with variable solar radiation, since both, $J(HONO)$ (sink) and $J(NO_2)$ (source) show a linear correlation. Thus, the argument is not valid.
Re: Thanks for your great comments. The following sentence and Figure S2 were removed in the revised manuscript and supplyment, respectively.
"On the other hand, the solar radiation intensity in this day (12 May, 2018) was significantly lower than other days (Fig. S2), and this weather condition was not conductive to the HONO formation through the heterogeneous reaction of $NO_2$."

Lines 191-192 and 205: Here very different $HONO/NO_2$ ratios are specified for the same (?) ship data? From the data in lines 191-192 I get values of 2.7 % and 2.4 % ("total averaged"), while in line 205 45 % are mentioned for the "average value"? Check data and/or explain differences.
Re: Thanks for your great comments. 0.027 and 0.024 were the average values of $HONO/NO_2$ at sea-oriented and land-oriented measurements during the whole campaign. 0.45 was the average value of $HONO/NO_2$ on 02, 12 and 14 May. The sentences have been rewritten as following:
"The surface concentration of $NO_2$ and HONO were extracted from their corresponding vertical profiles. As shown in Figure 3, the total averaged near-surface $NO_2$ concentrations under sea-oriented and land-oriented measurements were 8.46 and 11.31 ppb, respectively. The total averaged near-surface HONO concentrations were 0.23 and 0.27 ppb under sea-oriented and land-oriented measurements. Previous studies reported that vehicle and ship emissions were the main primary HONO sources on land and sea, respectively, and $NO_2$ heterogeneous reaction on the surfaces of ground, sea, vegetation and aerosol were the HONO important secondary sources (Liu et al., 2021). They also found that the surface HONO concentration under sea case was lower than that under land case, especially in the morning and evening (Yang et al., 2021). Figure 4 showed the time series of AOD, the surface concentrations of $NO_2$ and HONO, and the surface $HONO/NO_2$ during the whole campaign. We could find the time series of AOD and $NO_2$ were similar. The high AOD and $NO_2$ usually appeared in busy shipping channels and ports, and the obvious high-value areas were the coast of the Yangtze River Delta, the Taiwan Strait, Xiamen port, Zhanjiang port and Qingdao port (with mean AOD of 1.28 and mean $NO_2$ of 18.90 ppb). HONO always appeared under high AOD and $NO_2$ conditions, however, high AOD and $NO_2$ were not necessarily accompanied with high HONO concentration. This was because the heterogeneous formation of HONO requiring suitable meteorological conditions (i.e., RH and temperature) in addition to its

precursor (NO$_2$) and the reaction surface (aerosol) (Liu et al., 2019). The high HONO/NO$_2$ values were found on 02, 13 and 14 May with an average value of 0.45. Moreover, we found the high values of HONO/NO$_2$ always appeared from 11:00 to 14:00 during a whole day."

Line 202: should be "high HONO concentration". A production rate (dHONO/dt) was not determined and you may have a small production rate (slope) at high HONO.
Re: Thanks for your great comments. We have rewritten this sentence as following:
"HONO always appeared under high AOD and NO$_2$ conditions, however, high AOD and NO$_2$ were not necessarily accompanied with high HONO concentration."

Line 206-207: Check again the argument (see above, sources and sink scale with radiation…).
Re: Thanks for your great comments. The following sentence was removed in the revised manuscript.
"That was due to the high production rate of HONO and the high photolysis rate of NO$_2$ during noontime"

Section 3.2.1: Check whether the "turning points" (especially the two in Fig. 6c) are significant or just scatter of the data? In addition, possible "artificial correlations" should be discussed, see above.
And can you explain, why only the "six highest values" are shown in Fig. 6 (red data) and not the mean/median? Is that representative or are here only outliers shown?
Re: Thanks for your great comments.
In order to eliminate the influence of other factors, the average of six highest HONO/NO$_2$ in each 10% RH interval is calculated. The bands of RH were selected to be 40-50%, 50-60%, 60-70%, 70-80%, 80-90% and 90-100% in Figure 6 (c). In order to prove whether there was possibility of artificial correlation, we selected RH intervals of 5% (40-45%, 45-50%, 50-55%, 55-60%, 60-65%, 65-70%, 70-75%, 75-80%, 80-85%, 85-90%, 90-95% and 95-100%). We used mean HONO/NO$_2$ values during this process. In Figure R8, we could also find two turning peaks appearing at ~60% and ~85% (80-90%), respectively. As reported by Cui et al. (2019), it can also be found that two similar RH turning peaks corresponding to higher HONO/NO$_2$ values from the observation data in East China Sea, although they did not clearly explain this phenomenon in their manuscript.

[Figure]

Figure R8. Scatter plots of RH and HONO/NO$_2$ ratios in the ship-based campaign.

Line 245, 246, 251: Here continuously increasing or decreasing data is shown and the highest value are specified as "peak". However, the "peak values" were not determined and could be even at lower or higher temperatures…

Re: Thanks for your great comments. We have rewritten the sentences as following:

(1) "In inland condition (CAMS), the HONO/$NO_2$ decreased along with the increase of temperature, and the highest values of HONO/$NO_2$ appeared on ~12.5℃."

(2) "However, we found that HONO/$NO_2$ increased along with the increase of temperature, and the highest values of HONO/$NO_2$ appeared with ~31.5℃ in coastal condition (SUST)."

(3) "In sea condition, the HONO/$NO_2$ increased along with the increase of temperature with a high value under ~25.0℃ when the atmospheric temperature was larger than 18.0℃, simultaneously, a ~1.9 averaged HONO/$NO_2$ high value was found under ~15.0℃ (14.0-17.0℃)."

(4) "Moreover, we found that the appearance of HONO/$NO_2$ high values under lower temperature (14.0-17.0℃) usually accompanied by landing wind."

Paragraph lines 282-295/ figures 10 and 11: What is the difference between both figures? Seems to be the same? Define two cases?

Re: Thanks for your great comments. We would like to understand the difference of the vertical evolution of HONO/$NO_2$ under inland and sea scenes under different aerosol loads. Figure 10 introduced a case with low aerosol level (<0.2 km$^{-1}$) but with similar vertical shape of aerosol under inland and sea scenes. Figure 11 introduced a case with relatively high aerosol level (~0.8 km$^{-1}$) but with similar vertical shape of aerosol under inland and sea scenes.

*We have rewritten these sentences as following:*

"In addition, we selected inland cases (CAMS) to learn the difference of height dependence of HONO/$NO_2$ compared with sea scenes under different aerosol loads. As shown in Figure 10, the sea and inland scenes had the similar aerosol levels (low aerosol level: < 0.2 km$^{-1}$) and vertical structure. Moreover, the $NO_2$ and HONO in sea and inland scenes had the similar vertical structure, but their concentrations in sea scene are all larger than that in inland scene. In Figure 10(d), we could find that the HONO/$NO_2$ in sea scene was obviously larger than that in inland scene above 400 m. The HONO/$NO_2$ in sea scene was about 4.5 times larger than that in inland scene especially above 600 m. As shown in Figure 11, the aerosols under sea and inland scenes were also with the similar extinction levels (relatively high level: ~0.8 km$^{-1}$) and vertical structure. The $NO_2$ concentration in sea scene was higher than that in inland scene but with a similar vertical structure. The HONO concentration in sea scene was lower than that in inland scene under 400 m, while it in sea scene was larger than that in inland scene above 400 m. In Figure 11 (d), we found the HONO/$NO_2$ in inland scene was larger than that in sea scene under 600 m, while the HONO/$NO_2$ in sea scene was about 2 times larger than that in inland scene above 600 m. Above all cases indicated that the HONO generation rate from $NO_2$ heterogeneous reaction in sea scene was larger than that in inland scene in higher atmospheric layers above 400-600 m. The high-altitude (> 400-600 m) atmospheric parameters in sea scene were more conductive to promote the HONO formation through the heterogeneous reaction of $NO_2$."

Line 315: Where is that HONO peak at 12:15 in Figure 12c? I see a stronger peak at ca. 14:15…?

Re: Thanks for your great comments. There was a HONO peak at 12:15. In order to observe the data more intuitively, we plotted the HONO concentration at bottom layer on 20 April in Figure R9. We put this figure into the revised supplyment.

[Figure]

Figure R9. Time series of HONO at bottom layer on 20 April 2018.

Line 330-331: The two RH and especially the two T values are not very different to allow any conclusions to the mechanism.
Re: Thanks for your great comments. We have rewritten this sentence as following: "The slightly increase of RH and temperature (Tem) at 14:00-16:00 (RH: ~75.0%, Tem: 23.7℃) may contribute to HONO formation through heterogeneous reaction of $NO_2$ on the aerosol surface than that at 09:00-11:00."

Line 343-344, Fig. 15: Not the $NO_2$ concentration is increasing during this period (see color code), but the layer is getting thicker.
Re: Thanks for your great comments. As shown in Figure R10, we could find that $NO_2$ increased under 1.0 km from 08:00 to 12:00. We put this figure into the revised supplyment.

[Figure]

Figure R10. Time series of $NO_2$ at 6 layers on 03 May 2018.

Line 347-348. The peaks in HONO at ca. 9:45, 11:00, 11:45 and 12:30 in Fig. 15 are anticorrelated to $NO_2$ (in contrast to the statement…), which is very unusual? Check data and sentence.
Re: Thanks for your great comments. We checked the $NO_2$ and HONO data in this case, and the peaks of $NO_2$ and HONO at 0.5-1.0 km indeed appeared simultaneously from 09:45 to 13:00. We have rewritten this sentence as following:
"Several HONO peaks (> 0.2 ppb) at 0.5-1.0 km were found from 09:45 to 13:00, and the aerosol and $NO_2$ high values were also observed at this height layer, simultaneously."

Line 375: Should be "emission ratio".
Re: Thanks for your great comments. We have rewritten this sentence as following:

"In order to further understand the impacts of RH, temperature, and SRI on the heterogeneous reaction of $NO_2$ to produce HONO, the emission ratio of $\Delta HONO/\Delta NO_x$ in sea, inland and coastal areas were calculated with values of $0.46 \pm 0.31\%$, $0.82 \pm 0.34\%$, and $0.79 \pm 0.31\%$ to remove the primary HONO source."

Fig. 1: The data shown seems to be not "typical". The DSCDs in the figure are factors higher than the data described in section 3.1?
Re: Thanks for your great comments.
In section 3.1, we used VCD to depict the variation of $NO_2$ and HONO along the cruise route. The relationship between DSCD and VCD was $VCD = DSCD/DAMF$.
In this study, a radiative transfer model SCIATRAN was used to convert SCDs of $NO_2$ and HONO to their tropospheric vertical column densities (VCDs). The vertical profiles of aerosol, $NO_2$ and HONO retrieved from MAX-DOAS, the temperature and pressure vertical profiles simulated using a dynamical-chemical model (WRF-Chem), and the geo-position data collected by GPS were introduced as inputs in SCIATRAN for the $NO_2$ and HONO air mass factor (AMF) calculation.
We also provided the conversion relationship between DSCD and VCD based on geometric AMF to help you to quickly quantify this relationship.
$$VCD = DSCD\big/\big((1/\sin\alpha)-1\big).$$

$\alpha$ was the elevation angle. In actual observation, the real AMF (radiative transfer model based) will be larger than the geometric AMF, due to the multiple scattering effect of aerosols in the atmosphere.

Figure 3. Check the HONO/$NO_2$ data. I get 0.027 and 0.024 using the HONO (0.23 /0.27) and $NO_2$ (8.46/11.31) data?
Re: Thanks for your great comments. We have updated Figure 3 according to the actual observation data. The average HONO/$NO_2$ ratios in sea-oriented and land-oriented measurements should be 0.027 and 0.024, respectively.

[Figure]

Figure 3. Averaged aerosol extinction, $NO_2$ concentration, HONO concentration and HONO/$NO_2$ ratio during the campaign. The red and blue boxes denoted sea-oriented and land-oriented measurements, respectively.

Figure 6: please show the red/right y-axis scaling in all figures (will be different in a) and b)).
Re: Thanks for your good suggestion. We have replotted Figure 6 as following:

[revised manuscript text omitted]

---

## Author Comment (AC2)

Thank you for your careful review and constructive suggestions. These suggestions are quite valuable to us, and help improve our manuscript a lot.

**Point-to-point responses**

We appreciate the reviewers for their valuable and constructive comments, which are very helpful for the improvement of the manuscript. We have revised the manuscript carefully according to the reviewers' comments. We have addressed the reviewers' comments on a point-to-point basis as below for consideration, where the reviewers' comments are cited in **black**, and the responses are in **blue**.

Heterogeneous reaction of NO2 on wet surfaces is an important source of HONO. However, there are still many uncertainties in the research on the mechanism of the heterogeneous reaction of NO2 to produce HONO, and a complete consensus has not yet been reached in the scientific research community. Pseudo-steady-state calculations and model simulations also show that HONO levels will be greatly underestimated by considering only homogeneous chemical reactions. At present, the assessment of the contribution of the heterogeneous reaction of NO2 to HONO in the vertical boundary layer has not been fully determined, which hinders the in-depth understanding of the distribution characteristics of tropospheric HONO, the transformation and formation process and its environmental effects. In addition, the research on HONO and its precursors in coastal and offshore scenarios is not sufficient, resulting in a lack of understanding of the ocean-atmospheric nitrogen cycle and the sea-land-atmosphere interaction.

Xing et al. can not only provide data support for the improvement of atmospheric chemistry models, but also provide new insights for exploring the vertical sources of HONO on land and sea and the effect of photolysis on the oxidation capacity of the upper atmosphere, but also for the prevention and control of atmospheric composite pollution and PM2.5. The synergistic control with O3 provides new scientific basis and clues. I suggest publication in ACP after minor revision. The detailed comments are as follows:

1. In this study, the uncertainty evaluation is imperfect. I suggest the authors to add a section or even in the supplement to explain the uncertainties of data or how trustworthy of the presented data in this manuscript.

Re: Thanks for your great comments.

We have supplemented error analysis in the main text as follows. Main text:

**"2.4 Error analysis**

[revised manuscript text omitted]

2. The authors should explain the meaning of this works clearly. Moreover, I suggest to shorten the abstract, which is quite long and contains too many details.

Re: Thanks for your great comments. We have deleted many unnecessary details and further simplified the abstract as follows. And the meaning of this work has been emphasized by underlining.

"Ship based multi-axis differential optical absorption spectroscopy (MAX-DOAS) measurements were conducted along the marginal seas of China from 19 April to 16 May 2018 to measure the vertical profiles of aerosol, nitrogen dioxide (NO2), and nitrous acid (HONO). Along the cruise route, we found five hot spots with enhanced tropospheric NO2 VCDs in Yangtze River Delta, Taiwan straits, Guangzhou-Hong Kong-Macao Greater Bay areas, Zhanjiang Port, and Qingdao port. Enhanced HONO concentrations could usually be observed under high-level aerosol and NO2 conditions, whereas the reverse was not always the case. To understand the impacts of relative humidity (RH), temperature, and aerosol on the heterogeneous reaction of NO2 to form HONO in different scenes, the Chinese Academy of Meteorological Sciences (CAMS) and Southern University of Science and Technology (SUST) MAX-DOAS stations were selected as the inland and coastal cases, respectively. The RH turning points in CAMS and SUST cases were both  $\sim 65\%$  (60–70%), whereas two turning peaks (~60% and ~85%) of RH were found in the sea cases. As temperature increased, the HONO/NO2 ratio decreased with peak values appearing at ~12.5  $^{\circ}$ C in CAMS, whereas the HONO/NO2 gradually increased and reached peak values at  $\sim$ 31.5 °C in SUST. In the sea case, when the temperature exceeded  $18.0^{\circ}$ C, the HONO/NO2 ratio rose with increasing temperature and achieved its peak at ~25.0  $^{\circ}$ C. This indicated that high temperature can contribute to the secondary formation of HONO in the sea

atmosphere. In the inland case, the correlation analysis between HONO and aerosol in the near-surface layer showed that the ground surface is more crucial to the formation of HONO via the heterogeneous reaction of NO2; however, in the coastal and sea cases, the aerosol surface contributed more. Furthermore, we discovered that the conversion rate of NO2 to HONO through heterogeneous reaction in the sea case is larger than that in the inland case in higher atmospheric layers (> 600 m). Three typical events were selected to demonstrate three potential contributing factors of HONO production under marine conditions (i.e., transport, NO2 heterogeneous reaction, and unknown HONO source). This study elucidates the sea-land and vertical differences in the forming mechanism of HONO via the NO2 heterogeneous reaction and provides deep insights into tropospheric HONO distribution, transforming process, and environmental effects."

3. The methodology section is too simple, especially in the vertical profile inversion module. Authors should provide detailed descriptions even in supplement.

Re: Thanks for your great comments. We have supplemented some contents in Section 2.3 and Supplementary materials as follows.

Re: Thanks for your great comments. We have supplemented some contents in Section 2.3 and Supplementary materials as follows.

Section 2.3: "The detailed retrieval procedure is displayed in Appendix I and Figure S3."

Appendix I: "The maximum a posteriori state vector  $\mathbf{x}$  is determined by minimizing the following cost function  $\chi^2$ .

$$\chi^{2} = (\mathbf{y} - F(\mathbf{x}, \mathbf{b}))^{T} \mathbf{S}_{\varepsilon}^{-1} (\mathbf{y} - F(\mathbf{x}, \mathbf{b})) + (\mathbf{x} - \mathbf{x}_{a})^{T} \mathbf{S}_{a}^{-1} (\mathbf{x} - \mathbf{x}_{a})$$
(1)

Here,  $F(\mathbf{x}, \mathbf{b})$  is the forward model, which describes the measured DSCDs  $\mathbf{y}$  as a function of the retrieval state vector  $\mathbf{x}$  (i.e., aerosol and trace gas vertical profiles) and the meteorological parameters  $\mathbf{b}$  (e.g., atmospheric pressure and temperature profiles);  $\mathbf{x}_a$  denotes the a priori vector that serves as an additional constraint;  $\mathbf{S}_z$  and  $\mathbf{S}_a$  are the covariance matrices of  $\mathbf{y}$  and  $\mathbf{x}_a$ , respectively. The retrieval of vertical profiles of aerosols and trace gases were classified into two steps (Figure S3). First, we retrieved vertical aerosol profiles based on a series of retrieved O4 DSCDs at different elevation angles. Second, the retrieved aerosol extinction profiles were utilized as the input parameters to the RTM to retrieve NO2 and HONO vertical profiles. Each scanning sequence of DSCD results (~5.5 min) correspond to one retrieved vertical profile information. In this study, we separated the atmosphere into 19 layers from 0 to 3.8 km with a vertical resolution of 0.2 km. Given the low sensitivity of MAX-DOAS measurements to high altitude and low concentration of pollutants above 3.0 km, we only displayed the vertical profiles below 3.0 km in this work."

Figure S3. Flowchart of the aerosol and trace gas retrieval algorithm. The dashed-lined red boxes denote the retrieval steps: aerosol and trace gas profile retrieval.

4. Section 3.4: The case study is too subjective. The authors should add detailed reasons for the case selection. Furthermore, section 3.4.2 lacks of sufficient proof. Re: Thanks for your great comments. We added the reasons for the case selection in front of Section 3.4.1-3.4.3 as follows.

"The important factors and precursors to drive the formation of HONO through heterogeneous reaction had complex evolution and transport characteristics. To further clarify the role of these parameters in the heterogeneous process of  $NO_2$  to form HONO, three typical processes were selected to reveal the favorable conditions for HONO formation at sea scene."

To make demonstrations more reasonable, we have done the following revisions:

1) Supplement a figure about backward trajectories of air masses to support our discussion.

"NO2 was mainly distributed near the sea surface layer 0-200 m, and a high-concentration NO2 air mass was found from 1.0–2.0 km during 13:00–14:00 due to the short distance transport of NO2 emitted from ships in Xiamen port (Figure S13)."

Figure S13. Daily 72-h backward trajectories of air masses in Xiamen port at (a) 1000 m, (b) 1500 m, and (c) 2000 m on 28 April 2018, respectively.

2) Transform some decisive descriptions into inferential ones.

"The higher RH and temperature (Tem) (RH: ~75.0%, Tem:  $23.7^{\circ}$ C) at 14:00-16:00 than that (RH: ~67.6%, Tem:  $23.1^{\circ}$ C) at 09:00-11:00 (Figure 14

(d)-(e)) promoted the HONO formation from the heterogeneous reaction of NO2 on the aerosol surface during 14:00-16:00." -> "The slight increase of RH and temperature (Tem) at 14:00–16:00 (RH: ~75.0%, Tem:  $23.7^{\circ}$ C) may contribute to HONO formation through heterogeneous reaction of NO2 on the aerosol surface than that at 09:00–11:00 (Figure 13 (d)-(e), Section 3.2)."

3) Add corresponding citations to support demonstrations.

"The higher SRI accelerated the photolysis of HONO during 09:00-11:00 period" -> "The higher SRI accelerated the photolysis of HONO during 09:00-11:00 period (Kraus et al., 1998)."

5. The conclusion is too long and should be shorten. Moreover, the implication the ship-based observation should be also added.

Re: Thanks for your great comments. We have deleted unnecessary details in the conclusion and supplemented the implication of ship-based observations as follows.

"Currently, many uncertainties in the study of the HONO forming mechanism through the heterogeneous reaction of NO2 exist. Earlier studies mostly focused on the near-surface layer, and the assessment of the contribution of NO2 heterogeneous reaction to HONO formation in the vertical direction of the boundary layer is insufficient. Therefore, we aim to learn the sea-land and vertical differences of the HONO forming mechanism from NO2 heterogeneous reaction and provide deep insights into the distribution characteristics, transforming process, and environmental effects of tropospheric HONO. Ship based MAX-DOAS observations along the marginal seas of China were performed from 19 April to 16 May 2018. Simultaneously, two ground-based MAX-DOAS observations were conducted in the inland station CAMS and the coastal station SUST to measure the aerosol, NO2, and HONO vertical profiles.

Along the cruise route, we found five hot spots with enhanced tropospheric NO2 VCDs in Yangtze River Delta, Taiwan straits, Guangzhou-Hong Kong-Macao Greater Bay areas, Zhanjiang Port, and Qingdao port. Under high-level NO2 conditions in the above five hot spots, we also observed enhanced HONO levels. Contrastingly, the low-concentration HONO accompanied high-level NO2 in the southeast coastline of Jiangsu province. When peak AOD and NO2 conditions were observed, enhanced HONO were observed, although the reverse was not always the case.

To understand the impacts of RH, temperature, and aerosol on the heterogeneous reaction of NO2 to produce HONO, the emission ratios of  $\Delta$ HONO/ $\Delta$ NOx were calculated to quantify the contribution of the primary HONO source to the total production of HONO. We found that the RH turning points in CAMS and SUST cases were both ~65% (60–70%), whereas two turning peaks (~60% and ~85%) of RH were found in the sea cases. This implied that high RH could contribute to the secondary formation of HONO in sea atmosphere. With increase in temperature, the HONO/NO2 decreased with peak values appearing at ~12.5  $^{\circ}$ C in CAMS, whereas the HONO/NO2 gradually increased and reached peak values at ~31.5°C in SUST. In the sea case, when the temperature exceeded 18.0  $^{\circ}$ C, the HONO/NO2 increased with the increasing temperature and achieved peak at ~25.0  $^{\circ}$ C. This indicated that high temperature could promote the secondary formation of HONO in the sea and coastal atmosphere. Additionally, the correlation analysis under different sea-land conditions indicated that the ground surface is more crucial to the formation of HONO from NO2 heterogeneous reaction in the inland case, whereas the aerosol surface contributed more in the coastal and sea cases.

Furthermore, we found that the HONO/NO2 in the sea case was about 4.5 times larger than that in the inland case above 600 m when AEC was ~0.2 km-1, and the HONO/NO2 ratio in the sea case was about 2 times larger than that in the inland case above 600 m when AEC was ~0.8 km-1, which implied that the generation rate of HONO from NO2 heterogeneous reaction in the sea case is larger than that in the inland case in higher atmospheric layers (> 600 m). To have a deep understanding of three potential contributing factors of HONO production under marine condition, we selected three typical events, which represented the impacts of transport, NO2 heterogeneous reaction, and unknown HONO source, respectively.

---

## Author Comment (AC3)

*Thank you for your careful review and constructive suggestions. These suggestions are quite valuable to us, and help improve our manuscript a lot.*

Point-to-point responses

*We appreciate the reviewers for their valuable and constructive comments, which are very helpful for the improvement of the manuscript. We have revised the manuscript carefully according to the reviewers' comments. We have addressed the reviewers' comments on a point-to-point basis as below for consideration, where the reviewers' comments are cited in **black**, and the responses are in **blue**.*

A better understanding of the sources and formation mechanisms of HONO is important for understanding troposphere oxidation and processes of secondary pollution. Previous research has focused on the near-surface layer such that there is insufficient literature measuring heterogeneous formation of HONO in the vertical profile. This study uses MAX-DOAS to study the vertical distributions of HONO and its sources over the sea along a Chinese coastline and at coastal stations. Retrievals of vertical profiles of aerosol, NO2 and HONO allow the examination of the differences in heterogeneous production of HONO in sea versus inland cases.

This work can provide important new information on the variation of sources of HONO in the vertical profile within the lower troposphere. However, the evaluation of data uncertainty is incomplete and hinders the use of the study's findings. The detailed comments to be considered are below:

**Major Comments**

There is general lack of uncertainties or error estimates presented with the measurements throughout, which makes the significance of the findings and conclusions uncertain. Any comparison of averages should include the standard deviations (ex. the VCDs on line 6 and concentrations on line 9). The results and discussion section requires discussion of which results are statistically significant and, therefore, an important contribution to the knowledge of the field. Presenting the uncertainties associated with the retrieved vertical profiles of NO2 and HONO is also required to draw significant conclusions about trends (Lines 269 to 281 & figure 9). The optimal estimation method should have produced some estimate of error when retrieving the vertical profiles. These errors bars should be included in the figures.

Re: Thanks for your great comments.

We have supplemented error analysis in the main text as follows.

Main text:

"**2.4 Error analysis**

[revised manuscript text omitted]

We have added error bars in all the retrieval vertical profiles (i.e., Fig. 9-11) as follows.

[Figure]

Figure 9. Map (a) shows the two measurement points (A: black, sea-oriented with sea wind; B: red, land-oriented with land wind) during the campaign. Plots (b)–(e) show the vertical profiles of aerosol, NO2, HONO, and HONO/NO2 ratios in the above two measurement points, respectively.

[Figure]

Figure 10. Plots showing the vertical distributions of (a) aerosol extinction, (b) NO2 concentration, (c) HONO concentration, and (d) HONO/NO2 ratio. The blue and red lines represent a ship-based campaign case and a CAMS case, respectively.

[Figure]

Figure 11. Plots showing the vertical distributions of (a) aerosol extinction, (b) NO2 concentration, (c) HONO concentration, and (d) HONO/NO2 ratio. The blue and red lines represent a ship-based campaign case and a CAMS case, respectively.

Retrieval uncertainty is particularly important for the HONO/NO2 ratios due to error propagation. For example, since the sensitivity of the MAX-DOAS retrievals tend to decrease with increasing altitude, the ratio values at higher altitudes in the profile may be a function of the chosen a-priori values rather than the true state of the atmosphere, and therefore cannot be interpreted. In general, a discussion should be included on how the changing MAX-DOAS sensitivity with altitude impacts the shape and magnitude of the retrievals compared to the true atmosphere (either the methodology or results sections). Otherwise, the readers might assume that the MAX-DOAS vertical profiles are more accurate at higher elevations than is the case (versus, for example, the accuracy level of lidar vertical profiles of aerosol extinction). An example of a typical averaging kernel from the optimal estimation retrieval should thus be provided (ex. in the supplemental). Finally, what sensitivity testing was conducted in terms of the effect on the chosen a-priori shape on the shape of the retrieved profiles?

Re: Thanks for your great comments. We have added corresponding contents in main text and Supplementary materials as follows.

"In this study, an exponential decreasing a priori with a scale height of 1.0 km was used as the initial profile for both the aerosol and trace gases retrieval (Figure S4). The surface concentrations of aerosol, NO2, and HONO were set to 0.2 km$^{-1}$, 3.0 ppb, and 1.0 ppb, respectively. We assume a fix set of aerosol optical properties with asymmetry parameter of 0.69, a single scattering albedo of 0.90, and ground albedo of 0.05. Furthermore, the uncertainty of the aerosol and trace gases a priori profile was set to 100% and the correlation length was set to 0.5 km. The averaging kernels indicated that the sensitivity of the profile retrieval tended to decrease with increasing

altitude, and was especially sensitive to the layers within 0–1.5 km (Figure S5). The sum of the diagonal elements in the averaging kernel matrix is the degrees of freedom (DOF), which denotes the number of independent pieces of information contained in the measurements."

[Figure]

Figure S4. An example of the a priori and retrieved profiles from MAX-DOAS measurements in ship-based campaign (May 1, 2019 at 08:02 LT) for (a) aerosol extinction, (b) NO$_2$, and (c) HONO.

[Figure]

Figure S5. An example of averaging kernel results from MAX-DOAS measurements in ship-based campaign (May 1, 2019 at 08:02 LT) for (a) aerosol extinction, (b) NO$_2$, and (c) HONO.

The chosen a-priori profile shape really affects the shape of the retrieved profiles. In previous studies, we compared the results retrieved using four different a priori profiles (i.e., linearly, exponential, Boltzmann, and Gaussian profiles) and selected the most suitable one——Gaussian a priori profile (Xing et al., 2017). The aerosol extinction profile retrieved using the Gaussian a priori profile shows the best agreement with simultaneous lidar and balloon-based measurements.

We did a sensitivity test to estimate the uncertainty related to the choice of the a priori profile for retrieved results by varying the scaling height (either 0.5 km or 1 km) (Hendrick et al., 2014). We found that the relative change was within 30%.

Given the effects of the a priori profile, current studies mostly set a priori profiles based on the data from other sources (e.g., modelling and balloon-based in situ measurements). Therefore, it is significant to study and control the impacts of the a priori profile on the retrieved profiles. We plan to do improvements on this aspect in the future. The exact methods include estimating the surface concentrations and VCDs from the DSCDs at the lowest elevation angle and the highest elevation angle, respectively. This method can reduce the dependence of retrieval on other data and enhance the robustness of retrieval procedure.

2. The use of English needs some improvement. Typos and grammatical errors, such as missing the words "that" and "the" in many sentences, reduces overall clarity. The manuscript would benefit from professional English editing.

Re: Thanks for your important comments. We have let a language revision institution help us polish our language and correct mistakes.

[Figure]

**Minor Comments**

1. Lines 16 to 19. For improved English, these sentences should use the format "the HONO/NO2 ratio was observed to decrease with increasing temperature…"

Re: Thanks for your great comments.

"The $HONO/NO_2$ decrease along with the increase of temperature, and with peak values on ~12.5℃ in CAMS. The $HONO/NO_2$ increased along with increasing temperature, and with peak values on ~31.5℃ in SUST. In sea case, the $HONO/NO_2$ increased along with the increase of temperature with a peak value on ~25.0℃ under the temperature being larger than 18.0℃." -> "As temperature increased, the $HONO/NO_2$ ratio decreased with peak values appearing at ~12.5℃ in CAMS, whereas the $HONO/NO_2$ gradually increased and reached peak values at ~31.5℃ in SUST. In the sea case, when the temperature exceeded 18.0℃, the $HONO/NO_2$ ratio rose with increasing temperature and achieved its peak at ~25.0℃."

2. Lines 29-30. Under land or sea conditions?

Re: Thanks for your great comments. They are all under land conditions. If we don't add any special instructions, land conditions are the default.

3. Lines 34-35. Amend to "… nitrate amines that pose a threat to human health".
Re: Thanks for your great comments.
"In addition, as a nitrosating agent, HONO can produce carcinogenic nitrite amines to threat to human health (Zhang et al., 2015)." -> "Additionally, as a nitrosating agent, HONO can produce carcinogenic nitrite amines that pose a threat to human health (Zhang et al., 2015)."

4. Line 37. Suggest listing the important/known HONO formation reactions, similar to the introduction in the Wen et al. (2019) referenced.
Re: Thanks for your great comments. We have revised and changed our descriptions as follows.
"Photolysis of HONO in near ultraviolet bands (Eq. 1) is a substantial source of hydroxyl radicals (OH radicals), which are one of the most important oxidants in the tropospheric atmosphere."
"Currently, the known sources of HONO mainly include direct emissions from vehicles, ships, biomass burning and soil, the homogeneous reaction of NO and OH radicals (Eq. 2), the nighttime and daytime heterogeneous reaction of $NO_2$ (Eq. 3) on aerosols, vegetation, ground and other types of surfaces, and the photolysis of nitrate particles (Eq. 4) (Alicke et al., 2003; Stemmer et al., 2006; Indarto et al., 2012; Wang et al., 2015; Salgado and Rossi, 2002; Zhou et al., 2011)."

$$HONO + hv \rightarrow NO + \cdot OH (\lambda < 400 nm) \tag{1}$$

$$\cdot OH + NO + M \rightarrow HONO + M \tag{2}$$

$$2NO_2 + H_2O \rightarrow HONO + HNO_3 \tag{3}$$

$$HNO_3 / NO_3^- + hv \rightarrow HONO / NO_2^- + O \cdot (\lambda \sim 300 nm) \tag{4}$$

5. Lines 39-40. Should this be "there are sources of HONO that are poorly understood"? Rewrite for clarity.
Re: Thanks for your great comments.
"There are also some obvious unknown HONO sources (Fu et al., 2019)." -> "Sources of HONO exist that are poorly understood (Fu et al., 2019)."

6. Lines 68 – 69. Are "favourable weather conditions" sea breeze conditions? Otherwise, what does "favourable" mean here?
Re: Thanks for your great comments.
"The formed HONO is completely likely to be transported to land cities at night under favorable weather conditions." -> "The formed HONO is likely to be carried to land cities at night by sea breeze, which will affect the atmospheric oxidation and air quality, and even endanger human health."

7. Line 82. Add "above surface" after 120 m.
Re: Thanks for your great comments. We have added some cited literatures and changed descriptions.
"…and found the maximum value of HONO appeared at 120 m sourced from the heterogeneous reaction of $NO_2$ on aerosol surface under haze conditions." -> "Taking tower and aircraft as platforms, these techniques were performed to measure HONO vertical profiles, and it was found that the peak values of HONO usually appeared under 200 m at urban and suburban areas (Kleffmann et al., 2003; Stemmler et al.,

2006; Zhang et al., 2009; Wong et al., 2012; Meng et al., 2020; Zhang et al., 2020). These studies also revealed that the heterogeneous reaction of $NO_2$ on multiple surfaces (ground and aerosol etc.) was an important source of HONO under planetary boundary layer (PBL), especially in haze days. Furthermore, they also reported that the $HONO/NO_2$ ratios usually decreased with the increase of height under 200 m at inland and coastal areas."

8. Lines 145 – 152. Suggest adding more detail about the optimal estimation method. For example, that the aerosol vertical profiles are retrieved from the O4 DSCDs, which are then used as model inputs for retrieving trace gas vertical profiles. What was the magnitude of the a priori for the aerosol and trace gas retrievals? How many minutes of measurements were included in the retrieval of one profile?

Re: Thanks for your great comments. We have supplemented some contents in Section 2.3 and Supplementary materials as follows.

Section 2.3: "The detailed retrieval procedure is displayed in Appendix I and Figure S3."

Appendix I: "The maximum a posteriori state vector $\mathbf{x}$ is determined by minimizing the following cost function $\chi^2$.

$$\chi^2 = (\mathbf{y} - F(\mathbf{x}, \mathbf{b}))^T \mathbf{S}_\varepsilon^{-1} (\mathbf{y} - F(\mathbf{x}, \mathbf{b})) + (\mathbf{x} - \mathbf{x}_a)^T \mathbf{S}_a^{-1} (\mathbf{x} - \mathbf{x}_a)$$
(1)

Here, $F(\mathbf{x}, \mathbf{b})$ is the forward model, which describes the measured DSCDs $\mathbf{y}$ as a function of the retrieval state vector $\mathbf{x}$ (i.e., aerosol and trace gas vertical profiles) and the meteorological parameters $\mathbf{b}$ (e.g., atmospheric pressure and temperature profiles); $\mathbf{x}_a$ denotes the a priori vector that serves as an additional constraint; $\mathbf{S}_\varepsilon$ and $\mathbf{S}_a$ are the covariance matrices of $\mathbf{y}$ and $\mathbf{x}_a$, respectively. The retrieval of vertical profiles of aerosols and trace gases were classified into two steps (Figure S3). First, we retrieved vertical aerosol profiles based on a series of retrieved O4 DSCDs at different elevation angles. Second, the retrieved aerosol extinction profiles were utilized as the input parameters to the RTM to retrieve NO2 and HONO vertical profiles. Each scanning sequence of DSCD results (~5.5 min) correspond to one retrieved vertical profile information. In this study, we separated the atmosphere into 19 layers from 0 to 3.8 km with a vertical resolution of 0.2 km. Given the low sensitivity of MAX-DOAS measurements to high altitude and low concentration of pollutants above 3.0 km, we only displayed the vertical profiles below 3.0 km in this work."

[Figure]

Figure S3. Flowchart of the aerosol and trace gas retrieval algorithm. The dashed-lined red boxes denote the retrieval steps: aerosol and trace gas profile retrieval.

9. Lines 160 – 161. It says in section 3.1 that the radiative transfer model SCIATRAN was used to convert SCDs of NO2 and HONO to VCDs, but why were the vertical profiles retrieved using the optimal estimation method not used to calculate VCDs? This appears to be duplication of work. If there is a lack of confidence in the vertical profiles from the optimal estimation method, this should be explained. Were the VCDs calculated using the two different methods compared? If so, please include in the supplemental and justify the methodological choice.

Re: Thanks for your great comments. The SCDs was converted to VCDs using Eq. c1.

$$VCD = \frac{SCD}{AMF} \tag{c1}$$

Here, AMF can be simulated by radiative transfer model SCIATRAN, and then we got VCDs of each trace gas. We didn't calculate VCDs using profiles retrieved for two reasons below.

On one hand, one profile is retrieved from one scanning sequence of DSCDs, which contains 11 DSCDs result at different angles (i.e., 1º, 2º, 3º, 4º, 5º, 6º, 8º, 10º, 15º, 30º and 90º). In other words, we could only get one VCD from 11 SCDs if we calculate VCDs using retrieved profile. Comparatively, using simulated AMF to calculate VCDs can let us get one VCD from each SCD result, and we can get more data points along this cruise line. On the other hand, calculating VCDs from one profile means that we need to use concentration at each height to represent one layer's average concentration (Eq. c2), which would introduce larger uncertainties. And there are uncertainties in profile retrieval as well. Therefore, this indirect converting method will bring in many uncertainties, which largely affects the VCD results accuracy.

$$VCD = \sum_{i=0}^{20} (C_i \times H_i) \tag{c2}$$

We didn't calculate VCDs using the two different methods compared. Instead, we just use Eq. c1 to calculate VCDs for more data points and better data quality.

10. Line 172. Suggest changing "elevated" to "enhanced" to make it clear that the hotspots were much greater than background as opposed to elevated above the surface (i.e. "lifted").

Re: Thanks for your great comments.
"five elevated tropospheric $NO_2$ VCDs hot spots" -> "Five enhanced tropospheric $NO_2$ VCDs hot spots were observed…"

11. Lines 171 to 182. A bar chart comparing the averaged NO2 and HONO (with standard deviations) in the five areas will be useful for the reader in terms of observing differences in the distributions described in the text. Box and whisker plots might also be a good choice since they provide more details about outliers.

Re: Thanks for your great comments. We have added corresponding box plots in Supplementary materials as follows.
Main text: "The averaged $NO_2$ VCDs in above five areas reached $1.07 \times 10^{16}$, $1.30 \times 10^{16}$, $7.27 \times 10^{15}$, $5.34 \times 10^{15}$, and $3.12 \times 10^{15}$ molec. cm$^{-2}$, respectively (Figure S6(a)). HONO exhibited similar spatial distribution characteristics as $NO_2$, and the averaged HONO VCDs in above five hot-spot areas reached $1.01 \times 10^{15}$, $7.91 \times 10^{14}$, $6.02 \times 10^{14}$, $5.36 \times 10^{14}$, and $5.17 \times 10^{14}$ molec. cm$^{-2}$, respectively (Figure S6(b))."
Supplementary materials:

[Figure]

Figure S6. The VCD distribution of (a) $NO_2$ and (b) HONO for five high-level emission sources (i.e., the coastal areas of Yangtze River Delta, Taiwan straits, Guangzhou-Hong Kong-Macao Greater Bay areas, Zhanjiang Port, and Qingdao port) along the cruise.

12. Line 213. What is meant by "navigation areas"?

Re: Thanks for your great comments. The "navigation areas" meant the shipping routes and international ports. To avoid misunderstanding, we have changed this sentence as follows.
"Sun et al. (2020) reported that HONO concentrations could increase up to 40–100% over the navigation areas" -> "Sun et al. (2020) reported that HONO concentrations could increase up to 40–100% over the shipping routes and international ports…"

13. Lines 210 to 11. More details about these stations are required. Where they located (ex. latitude and longitude coordinates). What are the characteristics of the stations? For example, local pollution sources, topography, prevailing meteorological conditions, etc.

Re: Thanks for your great comments. We have added station information in Section 2.1 the measurement cruise and Supplementary materials as follows. The topography and distribution of two stations are obviously revealed in Figure S2.

Section 2.1: "To fully understand the differences of the impacts of RH, temperature, and aerosol on the HONO secondary formation in land and sea conditions, the Chinese Academy of Meteorological Sciences (CAMS) and Southern University of Science and Technology (SUST) MAX-DOAS stations were selected as inland and coastal areas for analysis, respectively. CAMS is located in the urban of Beijing (116.32ºE, 39.94ºN), and SUST is located in Shenzhen (114.00ºE, 22.60ºN) (Figure S2)."

Supplementary materials:

[Figure]

Figure S2. Cruise route and the location of two MAX-DOAS stations (CAMS and SUST).

14. Lines 216 to 222. Much more detail is needed in terms of how the emission ratios were determined (i.e. in the methodology section). Simply citing the literature is not sufficient.

Re: Thanks for your great comments. We have added some details and revised some descriptions as follows.

Main text: "By subtracting the average marine background of NO$_x$ and HONO from the ship plume emission values, the impact of background values is reduced and the emission ratio of ΔHONO/ΔNOx can be obtained, and this emission ratio can be used for quantifying the primary HONO (Sun et al., 2020; Xu et al., 2015). In this study, we used an averaged 0.46±0.31% emission ratio of ΔHONO/ΔNOx referring to Sun et

al. (2020) to understand the primary source of HONO on the sea surface during the campaign."

"Additionally, the calculation method of emission ratios of ΔHONO/ΔNOx in CAMS and SUST was referred from Xu et al. (2015), Liu et al. (2018), and Xing et al. (2021) (Appendix II)."

Supplementary materials:

"Appendix II: The criteria of the identification of fresh plumes

The fresh plumes were selected using the following criteria: (a) [$NO_x$]>40 ppb, (b) $NO/NO_x$>0.85, (c) good correlation performing between HONO and $NO_x$ (R>0.90), (d) short duration of plumes (<=2.0 h), and (e) $70^o$<SZA<$75^o$.

MAX-DOAS performed based on the collected solar scattering spectrum to retrieve aerosol, $NO_2$ and HONO. In general, we believed that the retrieved MAX-DOAS data was reliable, when SZA was not large than $75^o$. In order to reduce the influence of fast photolysis of HONO and $NO_2$, we usually selected data with $70^o$<SZA<$75^o$ to calculated $HONO/NO_x$ ratios from direct emission. In this condition, the photolysis rate of $NO_2$ was not large than $0.25 \times 10^{-3}$ $s^{-1}$."

15. Line 223. Since later sections show how meteorological variables impact the HONO/NO2 relationship, it would be helpful to add visualise the impact on these scatterplots using coloured marker points. For example, you could provide versions of these plots where each point has a color corresponding to the temperature. These plots may help to explain some of the outliers that are reducing the R values.

Re: Thanks for your great comments. We have added corresponding Figure in Supplementary materials and demonstrations in main text as follows.

Main text: "The corresponding temperature and RH conditions of each spot are displayed in Figure S8, which roughly reveals the impact of RH and temperature on the process of $NO_2$ forming HONO through heterogeneous reactions."

[Figure]

Figure S8. Scatter plots of HONO concentration vs. $NO_2$ concentration coloured by (1) relative humidity (RH) and (2) temperature in (a) CAMS, (b) SUST, and ship-based measurements of (c) sea-oriented and (d) land-oriented under static weather condition.

However, the meteorological data of CAMS and SUST stations only have the RH and temperature data of 08:00, 11:00 and 14:00. Thus, there are different numbers of data points in Figure S8 and Figure 5 in CAMS and SUST stations, which further trigger different R values.

16. Line 230. Please explain and justify this methodological choice in more detail.

Re: Thanks for your great comments. The reason why we selected the highest values instead of mean or median values is that the variation of highest values can display an overall varying range of spots. Nearly all the data spots are below the average highest values. And the area confined by highest values and x-axis can reflect concentration levels of data spots to some extent. For example, we found that the area determined by average highest value and x-axis was much larger in the ship-based campaign than in CAMS and SUST (Fig. 6), which indicated that the varying range of $HONO/NO2$ ratio was much larger in sea cases than in inland cases. To eliminate the influence of other factors, we took the average of six highest $HONO/NO_2$ instead of the highest ones. We revised the method descriptions as follows.

"The highest values can represent varying range of data in each interval and reveal concentration levels of data distribution. To eliminate the influence of other factors, the average of the six highest $HONO/NO_2$ in each 10% RH interval is calculated to reflect the distribution range of data in each interval (Liu et al., 2019). The dependence of the averaged top-6 $HONO/NO_2$ on RH reveal an overall variation tendency of $HONO/NO_2$ against RH."

17. Line 231. Define "turning points" for the reader.
Re: Thanks for your great comments.
"We found the RH turning points in inland (CAMS) and coastal (SUST) cases are all ~65% (60-70%)." -> "In the inland (CAMS) and coastal (SUST) cases, the RH turning points are both ~65% (60–70%), where increasing trend switches to decreasing tendency."

18. Lines 243 to 244. Please explain and justify the methodological choice of using the average of the six highest values.
Re: Thanks for your great comments. Given that the method we used is the same with Section 3.2.1, we simplified our descriptions here.
"Similar to the scatter plots of $HONO/NO_2$ against RH, we also adopted the averaged top-6 $HONO/NO_2$ values in each 5℃ interval to represent a general variation tendency of $HONO/NO_2$ against temperature."

19. Line 252. Does "landing winds" mean a land breeze?
Re: Yes, we have changed our descriptions here.
"Moreover, we found that the appearance of $HONO/NO_2$ peak values under lower temperature (14.0-17.0℃) usually accompanied by landing wind." -> "Furthermore, we found that the appearance of $HONO/NO_2$ high values under lower temperature (14.0–17.0℃) was usually accompanied by land breeze."

20. Lines 257-260. Given that the R squared of the inland correlation is so small (<0.1), the fitted slope cannot be reliably interpreted (has no statistically significant meeting).
Re: Thanks for your great comments. We have deleted the descriptions based on the correlation results of R values < 0.1, and replaced the original figure with box plots of $HONO/NO_2$ ratio under different aerosol extinction coefficient conditions as follows.

[Figure]

21. Lines 263 to 266. Showing the averages and standard deviations of the ratios and aerosol extinctions in a bar chart, perhaps in the supplementary section, would help the reader. This comparison will also help to determine whether the average values are statistically significantly different based on the standard deviations, which is important to justify your conclusions.

Re: Thanks for your great comments. We have supplemented standard deviations behind the average values and added a bar chart in Figure 8 as follows.

Main text: "Additionally, we found the averaged values of $HONO/NO_2$ were 0.011±0.004, 0.014±0.006, 0.008±0.003, and 0.007±0.003 when aerosol extinctions are 0–0.3, 0.3–0.6, 0.6–0.9 and > 0.9 km$^{-1}$ in the inland case, respectively (Figure 8(b)). As shown in Figure 8, the high values of $HONO/NO_2$ were mainly under aerosol extinction being less than 1.0 km$^{-1}$ with averaged values of 0.012±0.006 and 0.090±0.004 in the coastal and sea cases, respectively."

[Figure]

Figure 8. (a), (c), and (e) show the linear regression plots between surface aerosol extinction and HONO concentrations in CAMS, SUST and the ship-based campaign, respectively. Plots (b), (d), and (f) depicts the HONO/NO₂ ratio distribution under different aerosol extinction coefficient conditions in CAMS, SUST and the ship-based campaign.

22. Line 373. The sentence is vague. Consider revising to something like, "when peak AOD and NO2 conditions were observed, enhanced HONO were observed, but the reverse was not always the case."
Re: Thanks for your great comments.
"HONO always appeared under high AOD and NO₂ conditions." -> "When peak AOD and NO₂ conditions were observed, enhanced HONO were observed, although the reverse was not always the case."

23. Lines 374 to 376. Consider changing "to remove the primary HONO source" to "to quantify the contribution of the primary HONO source to the total production of HONO."
Re: Thanks for your great comments.

"the emission rates of $\Delta HONO/\Delta NO_x$ in sea, inland and coastal areas were calculated with values of 0.46±0.31%, 0.82±0.34%, and 0.79±0.31% to remove the primary HONO source" -> "To understand the impacts of RH, temperature, and aerosol on the heterogeneous reaction of NO₂ to produce HONO, the emission ratios of ΔHONO/ΔNOx were calculated to quantify the contribution of the primary HONO source to the total production of HONO."

24. Line 552. Figure 12 c). Suggest reducing the maximum value on the colour bar for the HONO concentrations to make the enhanced periods easier to see.

Re: Thanks for your great comments. Given that the demonstration based on Figure 12c is about HONO variation trend, we have added a line plot (Figure S12) to display the diurnal variation of HONO. We kept the same maximum value on the color bar among Figure 12, 13 and 14. In this way, it is more convenient to do a comparison of trace gas concentration levels on different days.

Main text: "The HONO was mainly distributed near the surface with a mean concentration of 0.07 ppb, and the two peaks were found in the early morning (averaged 0.15 ppb) and at 12:15 (averaged 0.11 ppb), respectively (Figure S12)."

Supplementary materials:

[Figure]

Figure S12. Time series of HONO at bottom layer on 20 April 2018.